# Inflammation mobilizes copper metabolism to promote colon tumorigenesis via an IL-17-STEAP4-XIAP axis

Yun Liao[1,2,9], Junjie Zhao[1,9], Katarzyna Bulek[1,3], Fangqiang Tang[1], Xing Chen[1], Gang Cai[1], Shang Jia [4], Paul L. Fox[5], Emina Huang [6], Theresa T. Pizarro[7], Matthew F. Kalady[6], Mark W. Jackson[7], Shideng Bao[6], Ganes C. Sen[1], George R. Stark[6], Christopher J. Chang[4,8] & Xiaoxia Li[1]✉

Copper levels are known to be elevated in inflamed and malignant tissues. But the mechanism underlying this selective enrichment has been elusive. In this study, we report a axis by which inflammatory cytokines, such as IL-17, drive cellular copper uptake via the induction of a metalloreductase, STEAP4. IL-17-induced elevated intracellular copper level leads to the activation of an E3-ligase, XIAP, which potentiates IL-17-induced NFκB activation and suppresses the caspase 3 activity. Importantly, this IL-17-induced STEAP4-dependent cellular copper uptake is critical for colon tumor formation in a murine model of colitis-associated tumorigenesis and STEAP4 expression correlates with IL-17 level and XIAP activation in human colon cancer. In summary, this study reveals a IL-17-STEAP4-XIAP axis through which the inflammatory response induces copper uptake, promoting colon tumorigenesis.

[1] Department of Inflammation and Immunity, Lerner Research Institute, Cleveland Clinic, Cleveland, OH 44195, USA. [2] Department of Laboratory Medicine, West China Hospital, Sichuan University, 610041 Chengdu, China. [3] Department of Immunology, Faculty of Biochemistry, Biophysics and Biotechnology, Jagiellonian University, 30-387 Krakow, Poland. [4] Departments of Chemistry and Molecular and Cell Biology, University of California, Berkeley, CA 94720, USA. [5] Department of Cellular and Molecular Medicine, Lerner Research Institute, Cleveland Clinic, Cleveland, OH 44195, USA. [6] Department of Cancer Biology, Lerner Research Institute, Cleveland Clinic, Cleveland, OH 44195, USA. [7] Department of Pathology, Case Western Reserve University, Cleveland, OH 44106, USA. [8] Howard Hughes Medical Institute, University of California, Berkeley, CA 94720, USA. [9] These authors contributed equally: Yun Liao, Junjie Zhao. ✉email: lix@ccf.org

Cancer cells adopt a unique metabolic program to meet the relentless demand for proliferation and survival[1]. The distinct metabolism is now recognized as a hallmark for cancer. Although tremendous strides have been made to understand the glucose and lipid metabolism of cancer cells, how heavy metals are utilized remains elusive. Early studies have repeatedly documented altered levels of select heavy metals in malignant tissue[2,3]. In particular, copper levels are elevated in sera and tissues from patients with different solid tumors, including colorectal cancer[4–11]. In addition to its role as cofactor for key metabolic enzymes, copper also directly contributes to tumor growth by serving as a cofactor for signaling molecules such as MEK1, which relays oncogenic BRAF signaling to ERK1/2[12–14]. Furthermore, copper chelation therapies have shown encouraging efficacy in a phase II clinical trial for late-stage triple-negative breast cancer[15]. This evidence collectively suggests a critical role for copper in the progression of cancer.

A fundamental question concerning the role of copper in cancer biology is how it is metabolized and selectively elevated in malignant tissue. Copper is primarily stored in the liver, which then releases it in a protein-bound form into the blood, and excess copper is secreted in the bile[16,17]. As a toxic heavy metal, copper levels are tightly regulated by a series of transporters, chaperones, and enzymes. Tissue cells take copper up from serum via the copper transporter Ctr1[18]. Ctr1 transports cuprous (Cu I) but not cupric copper (Cu II)[19]. This uptake mechanism entails that Ctr1 works in conjunction with a metallo-reductase when copper is available as Cu II. This two-step paradigm of copper uptake shows that cancer cells need to manipulate the expression of both transporter and reductase in order to regulate intracellular copper levels[16].

Intriguingly, the inflammatory response is the most well-characterized physiological process that mobilizes biological copper[20–22]. While the precise mechanism by which the immune response re-routes copper trafficking is unclear, it is well documented that copper accumulates in inflamed tissues and contributes critically to host defense against bacterial infection. On the other hand, chronic inflammation is also a well-established risk factor for cancer development, especially in the colon[23,24]. Several proinflammatory cytokines, including IL-17, IL-6 and IL-22[25–28], are well-established tumor-promoting factors for colon tumorigenesis and cancer progression. Thus, it is possible that inflammatory cytokines, which are now known to have profound and divergent potency in reprograming cell metabolism, may contribute to copper accumulation in cancer cells by upregulating machinery involved in copper uptake.

In this study, we report that STEAP4 is a key metallo-reductase that promotes cellular copper uptake in response to chronic inflammation during colon tumorigenesis. STEAP4 reduces cupric copper to the cuprous form[29,30]. We found that STEAP4 overexpression significantly increases biologically available copper in a colon cancer cell line, which correlates with enhanced metastatic potential. In addition, STEAP4 was highly induced in the inflamed colon, and deletion of STEAP4 from colonic epithelial cells impaired copper accumulation and tumorigenesis in the colon in a murine model of colitis-associated tumorigenesis. Furthermore, STEAP4 was readily induced by tumor-promoting cytokines such as IL-17, and the expression of STEAP4 was required for IL-17-induced elevation of copper uptake. Mechanistically, STEAP4-mediated copper uptake sustained IL-17-induced NFκB activation, while it suppressed caspase-3 activity by activating XIAP. Our study has identified STEAP4 as a regulatory node for inflammation-mediated cellular copper uptake and revealed a mechanism through which inflammation and copper promote tumor progression.

## Results

**IL-17 promotes copper uptake via the induction of STEAP4.** STEAP4 is unique among its family members in that its expression is regulated by proinflammatory cytokines[31]. Intriguingly, STEAP4, but not any other component of the copper uptake machinery, was readily induced in mouse colon organoids by the panel of cytokines we tested (Fig. 1a). Analysis of STEAP4 promoter sequence revealed binding sites for both NFκB and STAT3. To determine whether STEAP4 was transcriptionally induced by NFκB or STAT3, we analyzed the induction of STEAP4 expression by IL-17 (NFκB activator) and IL-22 (STAT3 activator). STEAP4 was induced within 4 h of IL-17 and IL-22 stimulation (Supplementary Fig. 1a). Interestingly, pretreatment with actinomycin D, a transcription inhibitor, abolished the induction of STEAP4 by IL-17 and IL-22, suggesting that STEAP4 expression might be transcriptionally regulated by NFκB and STAT3 activators. While IL-17 stimulation increased the level of STEAP4, CTR1 is constitutively expressed in the organoids (Fig. 1b). In vivo, IL-17 neutralization dramatically suppressed STEAP4 expression in the inflamed colon tissue from dextran sodium sulfate (DSS)-challenged mice (Supplementary Fig. 1b–d). The colocalization of STEAP4 with Ctr1 suggested that IL-17-induced STEAP4 expression might increase the cellular copper uptake. To test this possibility, we measured the activity of the superoxide dismutase 1 (SOD1), which utilizes copper as a cofactor to scavenge free superoxide[31] (Supplementary Fig. 1e), in primary colon epithelial organoids. In order to determine the rate of cellular copper uptake, we first treated the organoids with tetra-thiomolybdate (TTM), a cell permeable $Cu^{1+}$-specific chelator, to deplete the intracellular copper and thereby quench baseline SOD activity (Supplementary Fig. 1e). Afterwards, TTM was withdrawn from the culture to allow the recovery of intracellular copper level, which was monitored by measuring the SOD activity (Fig. 1c). Interestingly, IL-17-primed colon organoids exhibited more rapid recovery of SOD activity after TTM withdrawal, compared to unprimed organoids (Fig. 1c). Importantly, STEAP4 deficiency (St4 KO) significantly impaired the recovery of SOD activity, despite IL-17 priming, suggesting that STEAP4 is required for the IL-17-induced increase of copper uptake (Fig. 1d). To confirm the elevation of intracellular copper in response to IL-17 stimulation, we employed a small molecule-based fluorescent probe (CF4) that emits a fluorescent signal in the presence of cuprous copper cations. Live imaging using the CF4 probe indeed detected increased levels of cuprous ions in response to IL-17-stimulation in wild-type but not STEAP4-deficient cells and organoids (Fig. 1e, f). In addition, IL-17-induced copper uptake was ablated in Ctr1-deficient cells (Fig. 1g), suggesting that this process is dependent on the reductase activity of STEAP4 that generates $Cu^{1+}$ for Ctr1-mediated import. Finally, using atomic absorption spectroscopy (AAS), we confirmed that IL-17 stimulation induces cellular copper uptake into the cytoplasm in a STEAP4-dependent manner (Fig. 1h, i). Taken together, these data suggest that IL-17 drives copper uptake by upregulating STEAP4 expression, in conjunction with the copper transporter Ctr1.

**STEAP4-mediated copper uptake promotes tumor growth.** We sought to identify the biological function of the inflammation-driven cellular copper uptake. Given the well-documented role of IL-17 signaling in driving epithelial cell proliferation and survival in inflammation-associated tumorigenesis, we asked whether STEAP4-mediated copper uptake promotes tumor growth. To this end, we engineered an inducible STEAP4 expression system into the human colon cancer cell line Ls174t, which robustly expressed Flag-tagged STEAP4 upon doxycycline treatment (Fig. 2a). Consistent with previous reports[29,30],

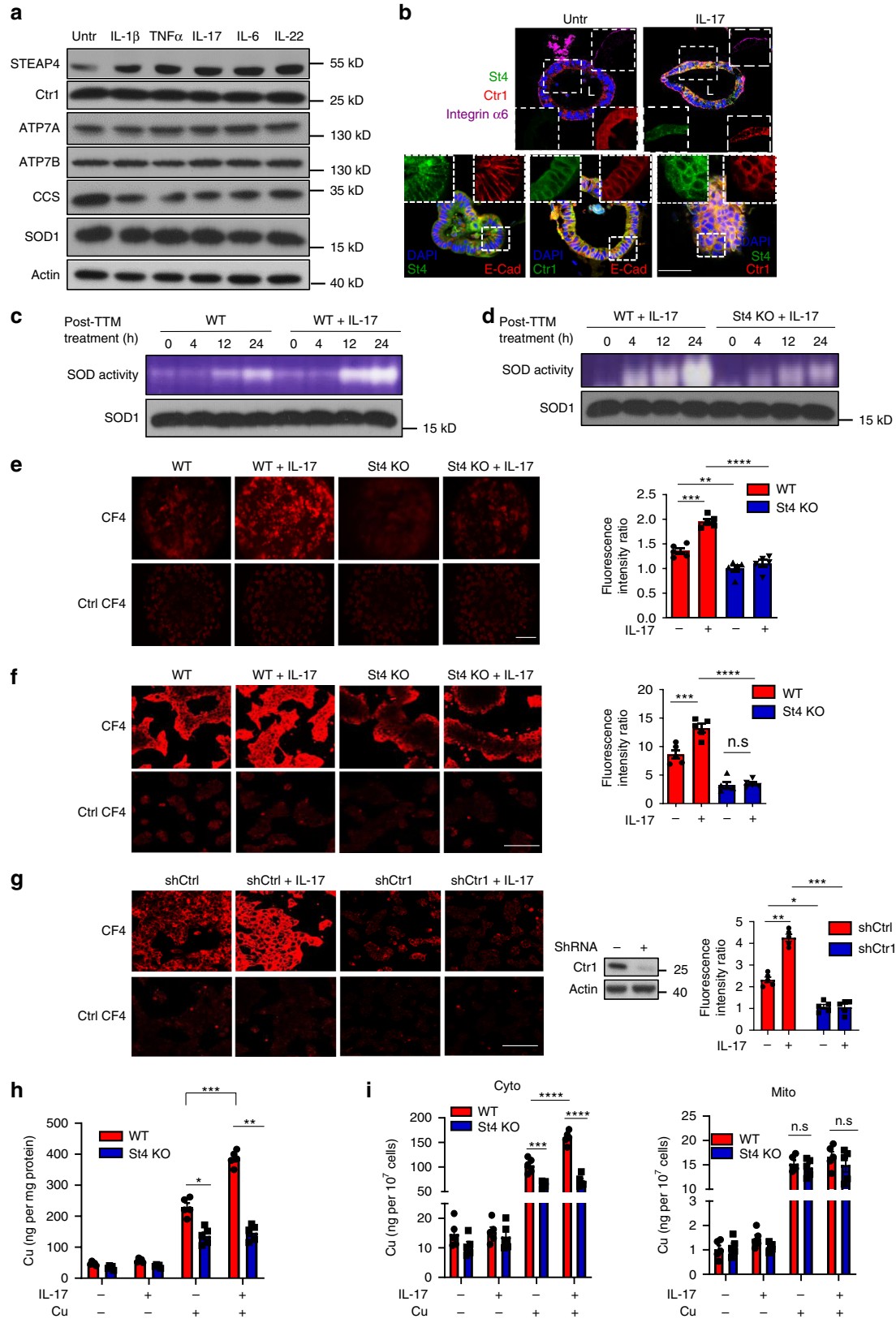

overexpression of STEAP4 increased total intracellular copper and iron pools in Ls174t cells (Fig. 2a). Interestingly, cellular fractionation showed that the increased intracellular copper in these cells was predominantly found in the cytoplasmic fraction, while the elevated iron was more abundant in the

mitochondrial fraction (Fig. 2b). In support of the elevated copper level in these cells, the activity of SOD1 was elevated in response to STEAP4 overexpression, suggesting that the increased intracellular copper was biologically functional (Fig. 2c). Consistently STEAP4-overexpressing cells exhibited

**Fig. 1 IL-17 induces cellular copper uptake through induction of STEAP4. a** Western blots analysis for copper trafficking-associated proteins from lysates of mouse colon epithelial organoids treated with different inflammatory cytokines for 8 h. Three independent experiments were done and representative blots were shown. Untr untreated. **b** IL-17 primed or unprimed mouse colon organoids were stained for STEAP4 (St4), E-cadherin (E-cad) and Ctr1 and integrin α. Scale bar, 100 μm. **c** SOD enzyme activity recovery from TTM inhibition in wild-type (WT) mouse organoids with or without 12 h of IL-17 priming. **d** SOD enzyme activity recovery in Steap4 wild-type (WT) and knockout (KO) organoids. **e**, **f** Fluorometric detection of intracellular copper in primary mouse colon organoids (**e**) or Ls174t cells (**f**). CF4 copper Fluor-4 probe, Ctrl CF4 control probe. Scale bar,100 μm. $n = 5$ biologically independent cell cultures. **g** Fluorometric detection of intracellular copper in Steap4 WT and Ctr1 knockdown cells. Cells were treated with or without IL-17 for 12 h and copper were supplemented into the medium for 6 h in all the groups. Scale bar,100 μm. $n = 5$ biologically independent cell cultures. **h** Atomic absorption spectroscopy measurement of copper content in Steap4 WT and KO Ls174t cells after copper addition. Cells were primed with or without IL-17 for 12 h and then 10 μM Cu(II) were supplemented into the medium of the indicated group for 6 h. $n = 5$ biologically independent cell cultures. **i** Copper content in cytosolic and mitochondrial fraction in Steap4 WT and KO cells, respectively. Cells were treated the same as described in panel (**h**). $n = 5$ biologically independent cell cultures. Results were normalized to cell numbers. All data show mean ± SEM. Two-tailed Student's $t$ test was performed for panel **e** (**\*\*$P = 0.0057$; \*\*\*$P = 0.0004$; \*\*\*\*$P < 0.0001$), **f** (\*\*\*$P = 0.0004$; \*\*\*\*$P < 0.0001$), **g** (\*$P = 0.015$; \*\*$P = 0.0027$; \*\*\*$P = 0.0004$), **h** (\*$P = 0.0383$; \*\*$P = 0.0022$; \*\*\*$P = 0.0009$), **i** (\*\*\*$P = 0.0003$; \*\*\*\*$P < 0.0001$; n.s., not significant). Data represent three independent experiments with similar results.

enhanced fluorescence when pulsed with active (CF4) but not a control probe (Ctrl CF4) (Fig. 2d).

We asked whether the increased level of copper could enhance tumor cell growth in vivo. We analyzed the growth kinetics of xenografts derived from Ls174t cells with and without STEAP4 overexpression using the STEAP4-inducible Ls174t cell clones. STEAP4 expression (Dox+) noticeably accelerated the growth of xenografts implanted in the flank of the NSG mice (Supplementary Fig. 2a). Furthermore, the growth of STEAP4-expressing tumors was suppressed by TTM treatment (Supplementary Fig. 2b). Immunofluorescence staining showed that the acceleration of tumor growth was associated with reduced caspase-3 activation and increased cell proliferation (Ki67+) in the tumor tissue, which was also dampened by copper chelation (Supplementary Fig. 2c, d). These data highlighted a critical role for STEAP4-mediated copper uptake in promoting tumor growth.

Next, we evaluated the impact of STEAP4 expression on the cancer metastasis and colonization in distant organ. We injected STEAP4-overexpressing (Dox+) or control (Dox−) cells into the spleens of immunocompromised mice to allow the formation of liver metastases (Fig. 2f–j). Mice injected with STEAP4-overexpressing Ls174t cells (Dox+, Supplementary Fig. 3a) showed reduced survival compared to mice injected with control cells (Fig. 2e). Consistently, in vivo bioluminescence and necropsy revealed increased liver metastases in mice receiving STEAP4-overexpresing cells compared to the controls (Fig. 2f, g). Notably, liver metastases derived from STEAP4-overexpressing Ls174t cells (Dox+) exhibited reduced apoptotic markers (cleaved caspase 3 and TUNEL) and comparable Ki67 staining (Fig. 2h). Importantly, TTM treatment reduced the burden of metastasis from STEAP4-overexpressing cells (Fig. 2I and Supplementary Fig. 3b, c). Additionally, TTM was able to further reduce the burden of metastases when administered in conjunction with a low dose of the chemotherapeutic agent 5-FU (Supplementary Fig. 3d). Taken together, the data indicate that STEAP4-mediated copper uptake plays a crucial role in promoting tumor growth.

**STEAP4 sustains NF-κB activation and inhibits apoptosis.** Given the profound impact of IL-17 stimulation on cellular copper uptake, we asked whether IL-17-induced STEAP4-mediated copper uptake might have any impact on downstream IL-17 signaling. To this end, we exploited the Ls174t cell line, which is highly responsive to IL-17 in serum-restricted cultures. Because DMEM does not contain copper and serum is the sole source of copper in normal cell culture, this condition allowed us to also assess the impact of copper deprivation and supplementation.

Consistent with increased intracellular copper levels, STEAP4 overexpression noticeably enhanced IL-17-stimulated ERK1/2 activation, which was completely subdued by copper chelation with TTM (Fig. 3a). Conversely, copper supplementation sustained IL-17-induced ERK1/2 in a STEAP4-dependent manner in Ls174t cells (Fig. 3b) as well as in primary mouse colon organoids (Fig. 3e). In addition, addition of copper also led to persisted activation of NFκB in response to IL-17 stimulation in a STEAP4-dependent manner (Fig. 3c, d and Supplementary Fig. 1f). Of note, NF-κB and ERK1/2 activation were comparable between wild-type and STEPA4-deficient cells at early time points in response to IL-17-stimulation (Fig. 3f), consistent with the dependence of copper-mediated enhancement of IL-17-induced NFκB and ERK1/2 activation on IL-17-induced STEAP4 expression.

In the intrasplenic injection model, STEAP4 overexpression was associated with reduced caspase-3 activation in liver metastases (Fig. 2h), and TTM/5-FU combined therapy induced dramatic remission of the tumors and increased caspase-3 activation (Supplementary Fig. 3d). To examine whether STEAP4-mediated copper uptake exerts a direct impact on caspase-3 activation, we examined the cellular response to 5-FU in the STEAP4-inducible colon cancer cell line and in STEAP4-deficient colon organoids. Consistent with our finding in the intrasplenic injection model, STEAP4 expression protected cultured Ls174t cells from 5-FU-induced caspase-3 activation (Fig. 3g). TTM treatment also restored the drug sensitivity (Fig. 3g). In contrast, IL-17 stimulation conferred upon both Ls174t cells and primary colon epithelial organoids resistance to 5-FU-induced caspase-3 activation in a STEAP4-dependent manner (Fig. 3h). Collectively, the data reveal that STEAP4-mediated copper uptake can suppress stress-induced caspase-3 activation.

Our mechanistic findings suggested that expression of STEAP4 might promote tumor growth via multiple downstream pathways. To assess the contribution of STEAP4-mediated suppression of caspase-3 activation to tumor growth, we used inhibitors against Bcl-2 family proteins as a tool to induce caspase-3 activation. We found that Ls174t cells were sensitive to s63845[32], which specifically inhibits Mcl-1. STEAP4 overexpression dampened s63845-mediated cell death at low concentrations (~100 nM) (Supplementary Fig. 4a). In vivo, although s63845 treatment suppressed the growth of both STEAP4-expressing (Dox+) and control (Dox−) xenografts, the s63845-treated STEAP4-expressing (Dox+) xenografts showed faster growth rate compared to control tumors treated with s63845 (Supplementary Fig. 4b, c). Consistently, immunofluorescence staining revealed that while s63845 treatment increased caspase-3 activation in both STEAP4-expressing xenograft tissue and control tumor tissue, STEAP4 expression was able to reduce the frequencies of cleaved caspase-3-positive cells in s63845-treated tumor (Supplementary Fig. 4d). The data suggested that STEAP4-mediated suppression of caspase 3 was downstream of chemical-induced caspase-3 activation.

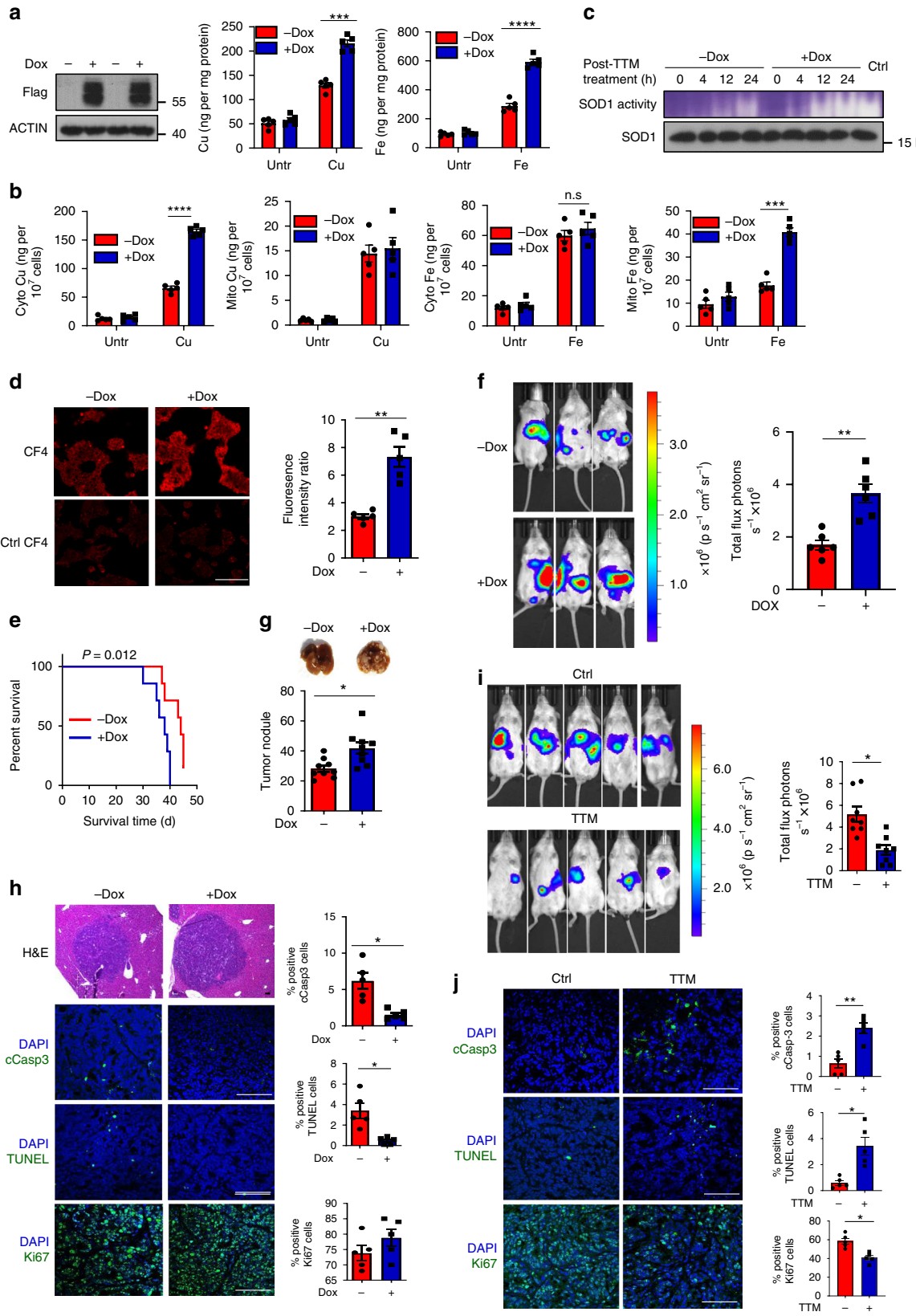

Interestingly, the baseline proliferative capacity of tumor cells in the xenograft from the flank model appeared to be lower than that in the liver metastasis (Fig. 2h and Supplementary Fig. 2c). Consequently, the proliferative impact of STEAP4 was more pronounced in the xenograft implanted in the flank (Supplementary Fig. 2c). A key question is which of the STEAP4-mediated

downstream signaling served to increase cell proliferation. Previous study has suggested that copper promotes oncogenic ERK1/2 signaling by potentiating the kinase activity of MEK1, a copper binding protein[13]. We indeed saw a copper-dependent enhancement in ERK1/2 signaling in response to IL-17 stimulation at late time points (Fig. 3a, b, e). Accordingly, treatment with

**Fig. 2 STEAP4-mediated copper uptake leads to tumor metastasis. a** AAS measurement of total copper and iron content in STEAP4-inducible Ls174t cells cultured in the presence and absence of doxycycline (Dox) in serum-free DMEM supplemented with 10 μM Cu(II) or 10 μM Fe(III). ***$P = 0.0009$; ****$P < 0.0001$ by two-tailed Student's $t$ test. $n = 5$ biologically independent cell cultures. **b** Copper and iron content in cytosol and mitochondrial fraction in STEAP4-inducible clone treated as described for panel (**a**). ***$P = 0.0009$; ****$P < 0.0001$; n.s., not significant by two-tailed Student's $t$ test. $n = 5$ biologically independent cell cultures. **c** Kinetics of the recovery of SOD activity from TTM treatment in STEAP4-inducible clones cultured in the presence or absence of Dox. Ctr, no TTM treatment. The experiment was repeated with 3 sets of biologically independent cell cultures. **d** Fluorometric detection of intracellular copper. CF4 copper Fluor-4 probe (CF4); Ctrl CF4 control probe. Scale bar, 100 μm. $n = 5$ biologically independent cell cultures. **$P = 0.0046$ by two-tailed Student's $t$ test. **e** Survival analysis for mice injected with STEAP4-inducible Ls174t clone cultured in the presence or absence of Dox ($P = 0.012$ by log-rank test, $n = 9$). **f** Metastatic tumor burden assessed 30 days after intrasplenic injection of luciferase-expressing STEAP4-inducible clones. Total flux photons were quantified in bar graph ($n = 6$ mice). **$P = 0.0017$ by two-tailed Student's $t$ test. **g** Total number of metastatic nodules per liver. *$P = 0.0151$ by two-tailed Student's $t$ test, $n = 9$ livers. **h** H&E and immunofluorescent staining of paraffin sections of the liver metastasis. Scale bar, 100 μm. ($n = 5$). *$P = 0.0158$ for Cleaved Caspase-3 (cCasp3) and $P = 0.0251$ for TUNEL by two-tailed Student's $t$ test. **i** Bioluminescence imaging of metastatic tumor burden after treatment with different drugs. Quantification was done based on the total flux photons. $n = 8$ mice. *$P = 0.0109$ by two-tailed Student's $t$ test. **j** Histology analysis for metastatic tumor tissues from mice receiving different treatments. Ki67, cleaved caspase-3 (cCasp3) were stained and TUNEL assay was performed on paraffin sections. $n = 5$ biologically independent tumor tissues. Scale bar, 100 μm. *$P = 0.0264$ for TUNEL, *$P = 0.0124$ for Ki67; **$P = 0.0040$ for cCasp3 by two-sided Student's $t$ test. All data were presented as mean ± SEM. Experiments in (**a**−**e**) were repeated three times; (**f**−**j**) were repeated twice. Data were not pooled.

the trametinib, a selective MEK1 inhibitor, suppressed STEAP4-mediated enhancement in cell proliferation in vitro (Supplementary Fig. 4e). In addition to the sustained ERK1/2 activation, STEAP4 expression also led to prolonged NFκB activation in a copper-dependent manner (Fig. 3c, d). We then examined the impact of STEAP4-mediated NFκB activation on tumor growth using a selective IKKα/β inhibitor BMS-345541[33,34]. Pharmacological inhibition of NFκB activation blunted STEAP4-induced cell proliferation in IL-17-stimulated Ls174t cells (Supplementary Fig. 4e). In line with the in vitro data, inhibition of NFκB activation reduced tumor cell proliferation and attenuated the growth advantage conferred by STEAP4 expression in the xenograft model (Supplementary Fig. 4f, g). However, inhibition of NFκB did not normalize the growth kinetics (Supplementary Fig. 4e, f). The residual tumor growth advantage was likely contributed by the STEAP4-mediated suppression of caspase-3 activation. Of note, IL-17-stimulated Ls174t cell line that expressed an enzymatically defective caspase-3 mutant (C163A) remained susceptible to TTM treatment-induced reduction in Ki67$^+$ cells and suppression of tumor growth (Supplementary Fig. 5a–g), highlighting a caspase-3-independent impact of STEAP4 on NFκB in promoting tumor growth.

**STEAP4-mediated copper uptake enhances the activity of XIAP.** While copper has been shown to directly impinge upon the activation of MAPK kinases via MEK1[13], the mechanisms by which copper regulates NF-κB and caspase-3 activation remain elusive. From the literature, we know that X-linked inhibitor of apoptosis (XIAP) is an E3 ligase that plays a critical role in mediating TAK1-dependent NF-κB activation[35]. XIAP also catalyzes the transfer of K48-linked ubiquitin moieties to caspase 3 to mediate its proteasome-dependent degradation[36,37]. Interestingly, a multitude of evidence suggests that the conformation of XIAP can be regulated by copper: supplementation to recombinant XIAP, that addition of copper to cell cultures induces a characteristic shift in XIAP mobility on nondenaturing gel[38], that addition of cuprous cations to recombinant XIAP protein leads to the formation of oligomers[39], and that copper can rescue zinc depletion-induced degradation of XIAP[40]. Furthermore, NMR analysis indicates that certain cysteine residues in XIAP are susceptible to copper-mediated oxidation, thereby regulating the oligomerization state of the protein[41]. These unique characteristics of XIAP prompted us to examine whether both the enhanced NF-κB activation and suppressed caspase-3 activity might be mediated by altered XIAP function, caused by an IL-17-induced, STEAP4-mediated increase in cellular copper uptake.

To test this hypothesis, we took advantage of the characteristic downward shift of XIAP mobility in response to copper supplementation in cell culture. We primed Ls174t cells with IL-17 to induce STEAP4 expression and cellular copper uptake in the presence of increasing doses of copper (II) and analyzed XIAP mobility by the Western method. Strikingly, IL-17-stimulation dramatically increased the mobility of XIAP protein (Fig. 4a). In addition, overexpression of STEAP4 in Ls174t cells also enhanced the downward shift of XIAP in response to copper (II) in a STEAP4-dependent manner (Fig. 4b, c). These data reveal that IL-17-induced STEAP4-dependent copper uptake has a major impact on the modification and/or conformation of XIAP. Importantly, copper supplementation failed to sustain IL-17-induced phosphorylation of IκB in XIAP-deficient cells (Fig. 4d), and XIAP deficiency rendered Ls174t cells highly sensitive to 5-FU-induced caspase-3 activation (Fig. 4e), indicating that XIAP plays an important role in the IL17-induced effects of copper in cancer cells. Importantly, Ls174t cell line that expressed a caspase-3 mutant (F381H) defective in the interaction with XIAP[42] (Supplementary Fig. 5h) exhibited elevated level of cell apoptosis (Supplementary Fig. 5i). While TTM treatment increased the TUNEL activity in IL-17-treated STEAP4-expressing cells with wild-type caspase 3, it failed to further enhance cell death in cells that expressed mutant caspase 3 (Supplementary Fig. 5i). Taken together, the data indicate that STEAP4-mediated suppression of caspase-3 activation was dependent on the XIAP−caspase-3 interaction. In sum, our xenograft data collectively (Supplementary Figs. 4–5) supported that the copper chelation-mediated reduction in tumor growth was due to both reduced proliferative output and increased apoptosis.

To examine whether copper-mediated XIAP-dependent signaling impact requires XIAP E3-ligase activity, we prepared recombinant XIAP protein in the presence of copper (XIAP-Cu) and in its absence. We then subjected the recombinant XIAP proteins to an in vitro ubiquitination assay, using cleaved caspase 3 as a substrate (Fig. 4f). XIAP-Cu exhibited greatly enhanced enzymatic activity compared to XIAP without copper. Previous studies have shown that the E3-ligase activity of XIAP promotes SOD1 activity via the ubiquitination of copper chaperone for superoxide dismutase (CCS)[43]. We indeed observed that XIAP deficiency reduced both basal and IL-17-induced SOD1 activity (Fig. 4g). Interestingly, we found that XIAP was found in a complex with SOD1 and CCS, and that this interaction was disrupted by STEAP4 overexpression (Figs. 4h, i). Taken together, the data reveal that IL-17-induced STEAP4 expression drives cellular copper uptake, leading to enhanced E3-ligase activity of XIAP. Increased XIAP E3-ligase activity results in a series of

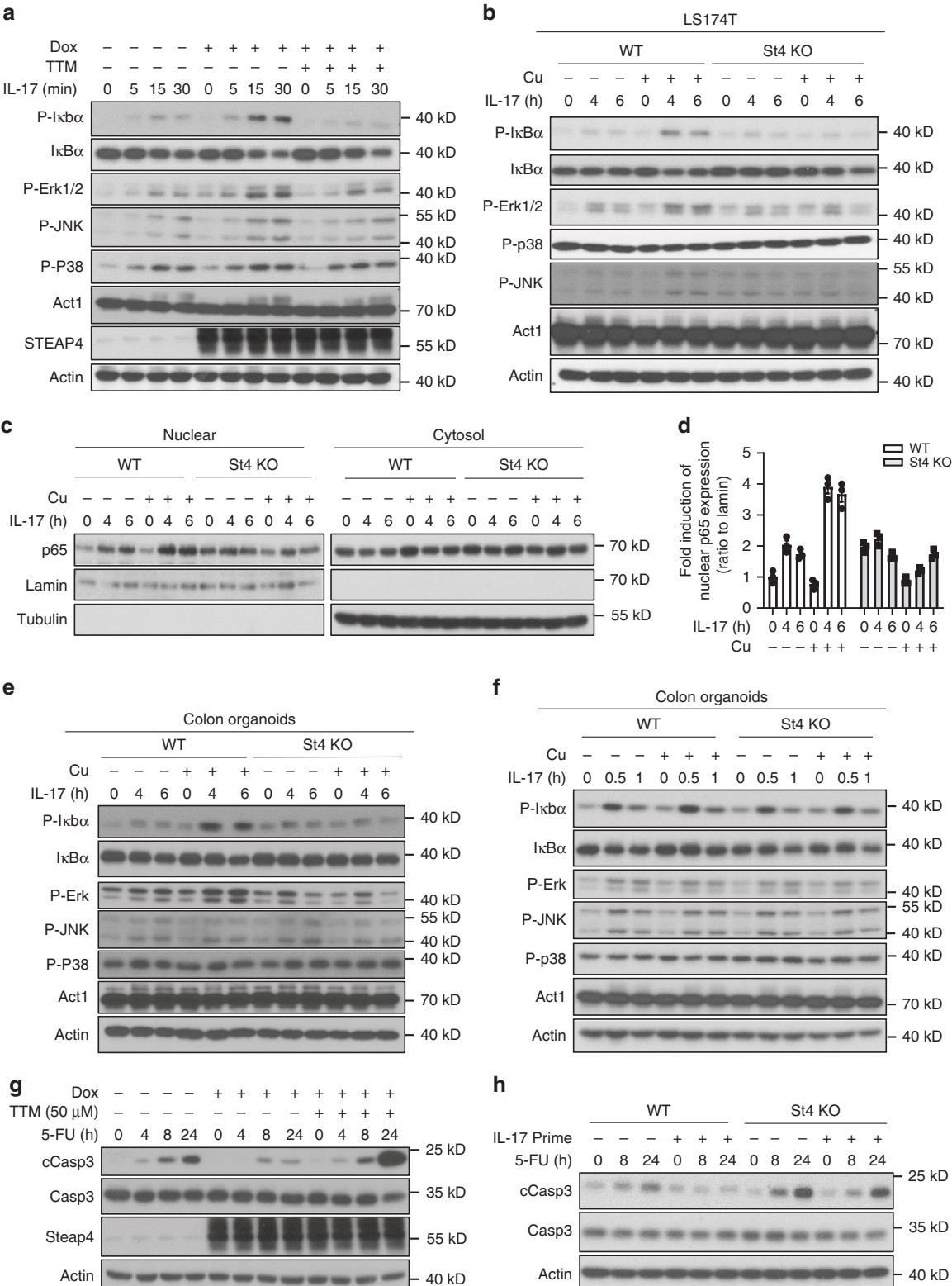

**Fig. 3 Copper uptake mediated by STEAP4 sustains NF-κB activation and inhibits apoptosis. a** Western blot analysis of STEAP4-inducible Ls174t cells stimulated with IL-17 in the presence or absence of Dox and/or TTM. **b** Western blot analysis of wild-type (WT) and STEAP4 knockout (St4 KO) Ls174t cells stimulated with IL-17 for indicated hours in the presence or absence of 10μesCu(II). **c** Western blot analysis for nuclear translocation of p65 in the presence or absence of TTM under IL-17 treatment in WT and St4 KO Ls174t cells stimulated with IL-17 for indicated hours in the presence or absence of 10μabCu(II). **d** Densitometric quantification of (**c**). Error bar shows S.E.M. of quantification from three biologically independent cells. **e, f** Western blot analysis of primary colon organoids from wild-type (WT) or STEAP4 knockout (St4 KO) mice stimulated with IL-17 for indicated hours in the presence or absence of Cu(II). **g** Western blot analysis showed Steap4 overexpression inhibited apoptosis, which can be reversed by TTM. **h** 5-FU-induced cell apoptosis in Steap4 WT and KO organoids primed or unprimed with IL-17. Unless specified otherwise, data represent three independent experiments with similar results.

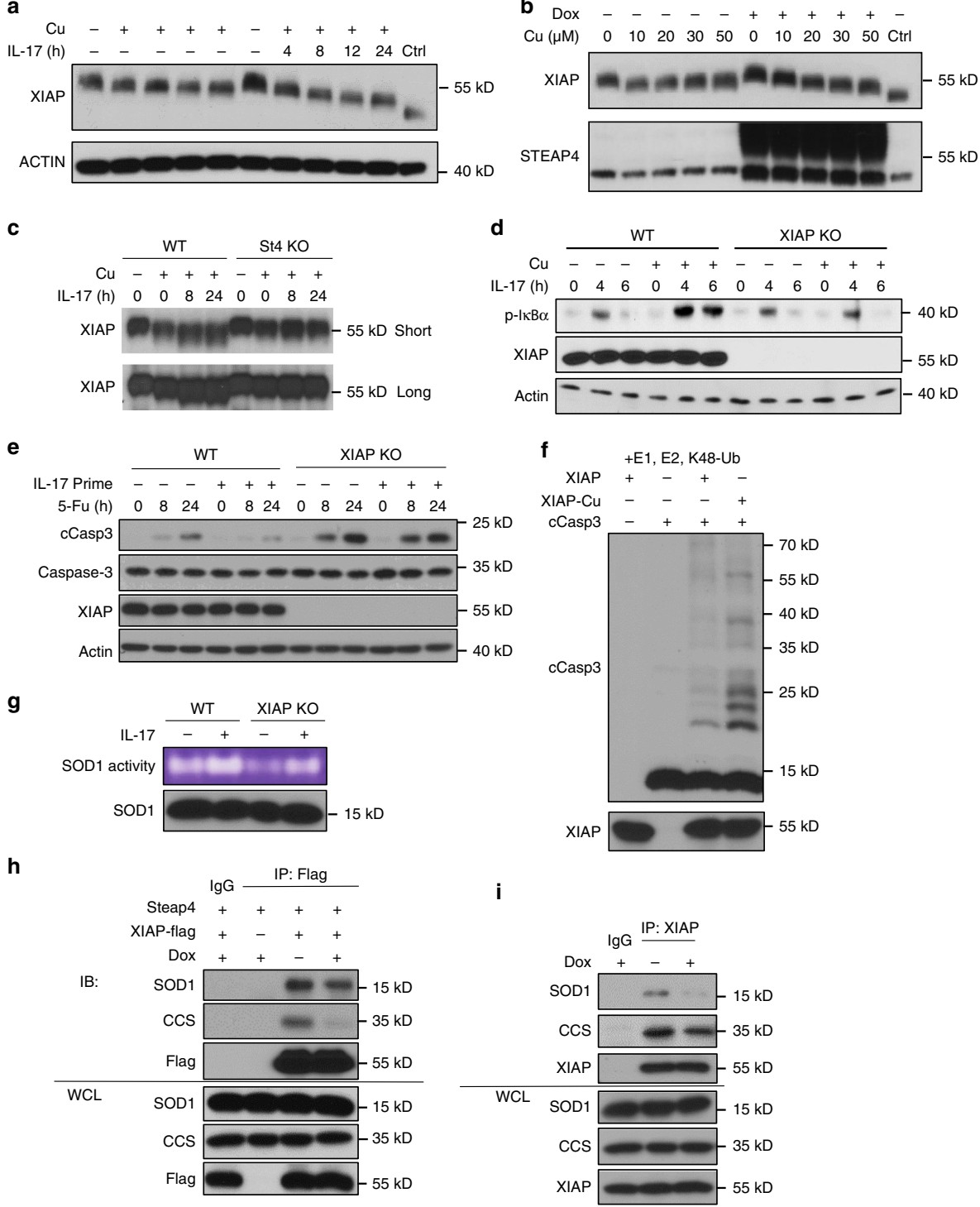

**Fig. 4 IL-17-induced STEAP4-mediated copper uptake enhances the E3-ligase activity of XIAP. a** IL-17 stimulation accentuates copper-induced XIAP shift. LS174T cells were pretreated with 10 μM Cu(II) for 24 h and then IL-17 was treated at the indicated time points. **b** STEAP4 overexpression accentuates XIAP shift. STEAP4 was induced by 1 μg/mL doxycycline for 24 h in the inducible clone and then treated with indicated concentrations of Cu (II) for 16 h. **c** STEAP4 is required for IL-17-induced accentuation of XIAP shift. Wild-type (WT) and STEAP4 knockout (St4 KO) intestinal organoids were first exposed to 10 μM Cu(II) for 16 h and then treated with IL-17 for indicated time points. **d** Western blots analysis for XIAP sustaining NF-κ signaling. XIAP WT and KO cells were pretreated with or without 10 μM copper, followed by IL-17 treatment for the indicated time points. **e** XIAP inhibits Caspase-3 cleavage. XIAP WT and KO cell line were first primed with IL-17 for 12 h and then treated with 5-FU for the indicated time points. **f** In vitro ubiquitination assay using recombinant XIAP prepared in the presence (XIAP-Cu) or absence (XIAP) of copper and cleaved human caspase-3 as substrate. **g** In-gel SOD activity assay in wild-type and XIAP knockout Ls174t in response to IL-17 stimulation. **h** Coimmunoprecipitation analysis of XIAP interaction with indicated protein in response to STEAP4 expression in STEAP4-inducible clone. **i** Coimmunoprecipitation analysis of endogenous XIAP in response to STEAP4 expression. Data represent three independent experiments with similar results.

IL-17-induced downstream responses, including sustained NF-κB activation, increased SOD1 activation, and suppressed caspase-3 activity.

**STEAP4 promotes colitis-associated colon tumorigenesis.** Previous studies have shown that STEAP4 is highly induced in colon epithelial cells in a mouse colitis model. Indeed, we found that DSS, widely used to trigger colitis in murine models, induced robust STEAP4 expression in mouse colon, starting from day 5 (Fig. 5a). Furthermore, the level of the copper chaperone CCS (Fig. 5a), which inversely correlates with the level of functional copper, was dramatically reduced as STEAP4 expression was increased. In addition, DSS treatment also induced the characteristic shift of the XIAP protein in colonic epithelial cells (Fig. 5b). Collectively, the data indicate that there is a concerted influx of copper in the mouse colon during intestinal inflammation. IL-17 has been shown to play a protective role on the integrity of intestinal epithelium by promoting tissue repair in response to DSS challenge[44,45,27] (Supplemental Fig. 1b–d). In support of the role of STEAP4 as an IL-17 target gene, colonic epithelial cell-specific STEAP4 deficiency resulted in aggravated disease in response to DSS challenged in terms of weight loss and increased gut permeability (Supplementary Fig. 6a–c). The exacerbation of disease was associated with reduced gut epithelial cell proliferation (Supplementary Fig. 6d). In light of the impact of STEAP4 expression on tumor growth, we sought to determine the role of STEAP4 in mouse colon epithelial cells in the colitis-associated cancer model. To this end, we generated conditional Steap4 knockout mice by flanking exon2 and exon 3 with two loxp sequences. By breeding this strain onto a CDX2-CreERT2 background, we derived colonic epithelial cell-specific Steap4 knockout mice (Fig. 5c, Steap4CKO). We then subjected the Steap4CKO and littermate control mice to chemically induced tumorigenesis (the AOM-DSS model). Steap4 deficiency resulted in pronounced reduction of copper and iron accumulation in enriched colonic epithelial cells (Fig. 5d, e), and the tumor burden in the Steap4CKO mice was also reduced compared to their littermate controls (Fig. 5f, g). Immunofluorescent analysis of the AOM-DSS-induced tumors showed that STEAP4 deficiency resulted in a reduced frequency of Ki67-positive and an increased frequency of active caspase-3-positive tumor cells (Fig. 5h). To determine whether increased copper uptake was required for AOM-DSS-induced tumorigenesis, we administered copper chelation therapy to wild-type mice treated with AOM-DSS (Fig. 5i). Therapy with TTM substantially reduced the final tumor burden in the AOM-DSS model (Fig. 5j, k). Similar to tumors from Steap4CKO mice, tumors from TTM-treated mice also showed reduced Ki67 staining and increased caspase-3 activation (Fig. 5l), indicating that copper influx into the colon is required for colitis-promoted tumorigenesis.

**IL-17-induced copper uptake contributes to chemoresistance.** Next, we sought to determine whether IL-17-induced STEAP4-mediated cellular copper uptake contributes to colon cancer progression in human. Consistent with a previous report[46], we found that STEAP4 was overexpressed in a subset of human colon cancer tissues and correlated with worse prognosis for human CRC (Fig. 6a and Supplementary Fig. 6e). Interestingly, the expression of STEAP4 positively correlated with IL-17 staining in the same tissues (Fig. 6b), while it inversely correlated with caspase-3 activation (Fig. 6c). This evidence led us to think that IL-17-induced STEAP4-mediated copper uptake might play an important role in sustaining tumor growth in response to chemotherapeutic treatments. In support of this idea, we found

that the XIAP protein from colon cancer tissues but not normal colonic epithelial cells exhibited the characteristic downward shift in nondenaturing gels (Fig. 6d). Of note, the increased XIAP mobility was detected in both primary cancers and in hepatic metastases, indicating that the level of biologically functional copper was higher in the malignant cells. Furthermore, IL-17 treatment activated copper uptake in cancer organoids generated from patient-derived liver metastases (Fig. 6e). In addition, IL-17-induced SOD1 activity and NF-κB activation were abolished by TTM or an XIAP inhibitor (Fig. 6f). IL-17 signaling has been previously shown to confer chemoresistance in colon cancer cells. We have now found that IL-17-mediated resistance to the caspase-3 activation that is induced by 5-FU was effectively abolished by copper chelation with TTM or by addition of XIAP inhibitor in patient-derived colon cancer organoids (Fig. 6g, h). Taken together, the data show that IL-17-induced copper uptake contributes to chemoresistance in human colon cancer.

**Discussion**
In this study, we provide genetic evidence supporting a critical role of STEAP4 in cellular copper transport in response to chronic inflammation during colon tumorigenesis. Deletion of STEAP4 specifically from colonic epithelial cells significantly reduces copper accumulation and attenuates colon tumorigenesis in the AOM-DSS model. Conversely, copper chelation abolishes enhanced hepatic colonization by colon cancer cells that over-express STEAP4. Mechanistically, STEAP4 expression enhances cellular copper uptake, activating the E3-ligase activity of XIAP, resulting in sustained NF-κB activation, increased SOD1 maturation, and suppressed caspase-3 activity. In summary, this study has identified a copper-dependent IL-17−STEAP4 axis that promotes cancer growth and confers resistance to chemotherapy.

Although the metallo-reductase activity of STEAP4 has been extensively characterized in vitro[29–31], the pathophysiological function of STEAP4 has remained elusive. A recent study from Xue et al.[46] has shown that STEAP4 overexpression in colonic epithelia cells increased the amount of iron in mitochondria at steady state and rendered mice susceptible to chemically induced colitis and colon tumorigenesis. By employing a loss-of-function approach, we found that both copper and iron accumulation were impaired in the colonic epithelial cell-specific STEAP4-deficient mice during the process of colon tumorigenesis. Importantly, copper chelation abolished STEAP4-mediated cancer growth in the xenograft model. In addition, copper chelation was sufficient to suppress tumorigenesis in the AOM-DSS model. Consistent with reduced copper accumulation in colon tissue, colonic epithelial cell-specific deletion of STEAP4 significantly attenuated colon tumorigenesis in the AOM-DSS model. Taken together, these data demonstrate a critical role for STEAP4-mediated copper uptake in tumor initiation and cancer progression.

Of note, STEAP4 is induced by inflammatory cytokines, including IL-17. STEAP4 deficiency abolished IL-17-induced copper uptake in cells, suggesting that STEAP4 expression is required for inflammation-mediated mobilization of copper metabolism. While previous studies have suggested that the mobilized copper is utilized by immune cells to kill bacteria during infection[47,48], our study reveals a crucial role for increased copper influx as a promoter of the survival of epithelial cells in response to inflammation. We identified a copper-regulated protein, XIAP, as a critical mediator linking cell survival and copper uptake. IL-17-induced incorporation of copper into XIAP led to enhanced E3-ligase activity, resulting in sustained NF-κB activation, increased SOD1 activity, and suppressed caspase-3 activation. These downstream responses collectively contribute to the survival of tumor cells during

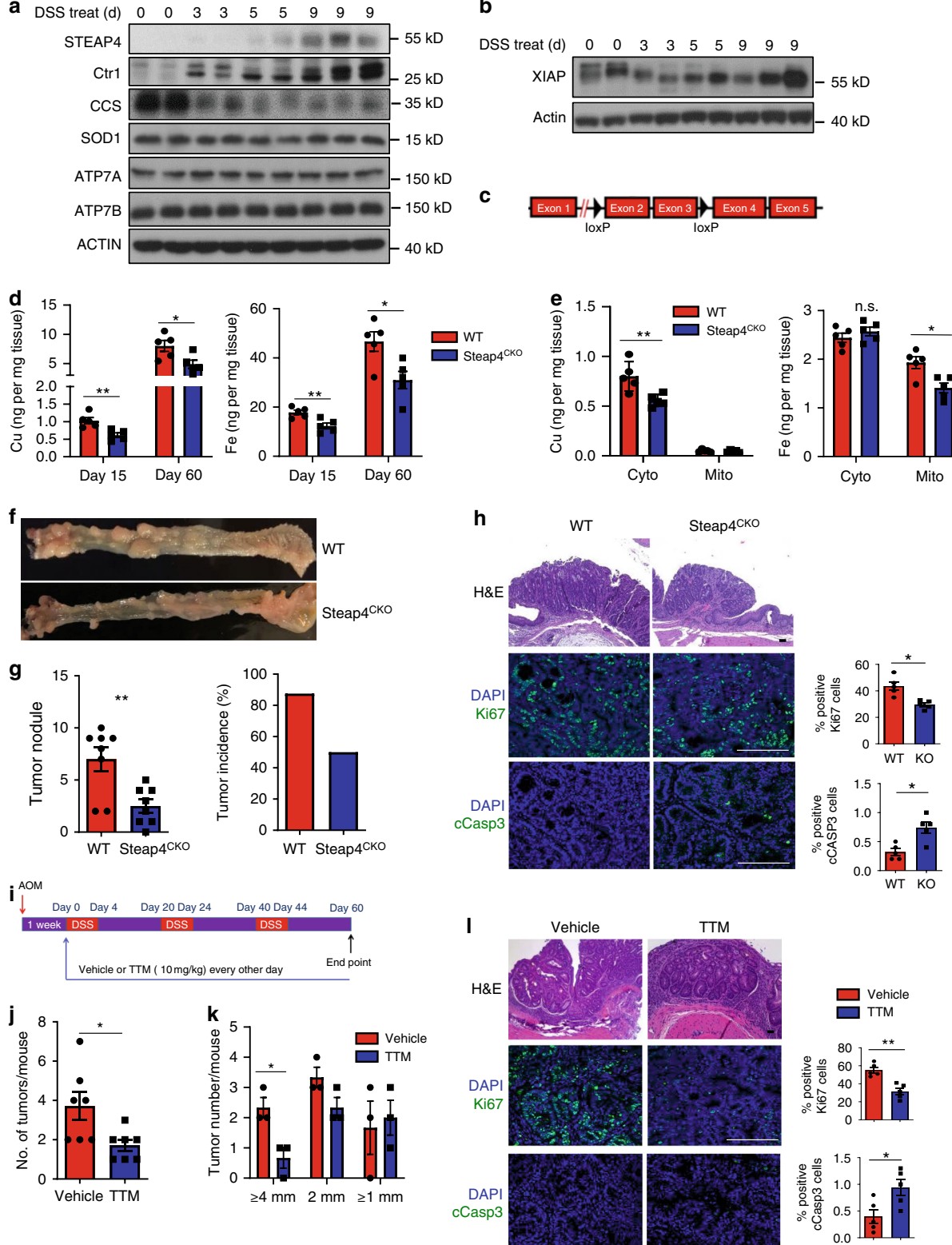

inflammation-associated colon tumorigenesis and confer resistance to chemotherapy-induced cell death.

The impact of copper on XIAP was first reported as an increase in the electrophoretic mobility of XIAP protein[38]. We found that IL-17 stimulation resulted in further enhancement of the shift in mobility of XIAP when copper was supplemented in cell culture media, indicating that increased cellular copper uptake can regulate XIAP modification and/or conformation. Of note, detection

of this characteristic shift required resolving protein lysates in gradient polyacrylamide gels in Bis−Tris buffer (Materials and methods)[38,43]. Two potential mechanisms could lead to the copper-dependent increase of electrophoretic mobility. Because of the multiple metal binding motifs in the XIAP protein[40], copper might bind to the protein directly, resulting in a conformational change. However, structural studies have yet to demonstrate the architecture of copper-bound XIAP. On the other hand, several

**Fig. 5 STEAP4 promotes colitis-associated colon tumorigenesis. a** Western blot analysis of colon tissue lysates from mice treated with DSS water for indicated days. **b** Analysis of XIAP shift on colon tissue lysates from mice treated with DSS water for indicated days. **c** Construction of STEAP4 floxed/ floxed allele. **d** Copper and iron contents of colons from WT and colonic epithelial cell-specific knockout mice (Steap4$^{CKO}$) treated with DSS. $n = 5$ mice. *$P = 0.0281$ and **$P = 0.0061$ for Cu measurement; *$P = 0.0186$ and **$P = 0.0063$ for Fe measurement by two-tailed Student's $t$ test. **e** Copper and iron contents in the cytosol fraction and mitochondrial fraction from the DSS-treated Steap4 WT and KO colon tissue. $n = 5$ mice. *$P = 0.0107$ for right panel; **$P = 0.0097$ for left panel by two-tailed Student's $t$ test. **f** Macroscopic view of longitudinally opened colon from control (WT) and colon epithelial cells-specific knockout mice. **g** Tumor numbers and tumor incidence from WT and Steap4$^{CKO}$ mice on AOM-DSS-induced colon cancer model. $n = 8$ mice. **$P = 0.0084$ by two-tailed Student's $t$ test. **h** H&E staining and immunostaining for Ki67 and cleaved caspase-3 (cCasp3) from paraffin sections of tumors of the indicated genotypes. Scale bar, 100 μm. $n = 5$ biologically independent tissues. *$P = 0.0239$ for Ki67; *$P = 0.0454$ for cCasp3 by two-tailed Student's $t$ test. **i** Treatment schedule of TTM. **j, k** Tumor burden for TTM- or vehicle-treated Steap4 WT mice on AOM-DSS model. *$P = 0.0232$ for panel (**j**) ($n = 7$ mice); *$P = 0.0241$ for panel (**k**) ($n = 3$) by two-tailed Student's $t$ test. **l** Immunostaining of Ki67 and cleaved Caspase-3 from TTM- or vehicle-treated colon tissue. $n = 5$ biologically independent tissues. *$P = 0.0265$ for cCasp3; **$P = 0.0012$ for Ki67 by two-tailed Student's $t$ test. Scale bar, 100 μm. All data were shown as mean ± SEM. All experiments were repeated twice with consistent results. Data were not pooled.

reports have shown that the BIR domains of XIAP contain multiple cysteine residues that are susceptible to copper-regulated oxidation[41,39], which could also account for the change in electrophoretic mobility. Although the nature of copper-induced XIAP shift remains unclear, we found that recombinant XIAP expressed in the presence of copper exhibited enhanced E3-ligase activity, indicating that copper can functionally impact XIAP. Future studies are required to elucidate the mechanism for copper-mediated regulation of XIAP structure and function.

An important question is how enhanced E3-ligase activity of XIAP promotes these distinct downstream responses. XIAP is a versatile protein with diverse roles in regulating cell death and inflammatory response. While well known for its ability to activate NF-κB signaling[35], XIAP is also an established E3-ligase for the K48 ubiquitination of cleaved caspase 3[36,37]. We found that copper-bound XIAP exhibited enhanced E3-ligase activity for cleaved caspase 3, which was abolished by copper chelation. Furthermore, XIAP has also been reported to promote the maturation of SOD1 activity via the ubiquitination of the copper chaperone CCS. We detected an XIAP-SOD-CCS complex that was regulated by STEAP4 expression. Considering the enhanced E3-ligase activity of copper-bound XIAP in vitro, it is possible that a STEAP4-mediated increase of cytosolic copper may trigger the potentiation of XIAP-E3-ligase activity, resulting in enhanced CCS ubiquitination and SOD1 maturation. Indeed, IL-17-induced SOD1 activity was greatly reduced in STEAP4- or XIAP-deficient cells. These data collectively support a central role for XIAP as the mediator of downstream responses of the IL-17−STEAP4 axis.

It is important to note that STEAP4- or XIAP-deficiency had no impact on the early-signaling responses induced by IL-17. Rather, STEAP4 and XIAP are required for copper-dependent sustained NF-κF activation in response to IL-17 stimulation. Importantly, the E3-ligase activity of XIAP has been reported to mediate the NF-κB activation that is induced by NOD activation by ubiquitinating RIPK2[49,50]. This sustained NF-κB activation has important pathological implications. IL-17 is a weak activator of NF-κB in cell culture models, which is in stark contrast to the profound proinflammatory impact of IL-17 in vivo. Our data suggest that the status of cellular copper metabolism may be a critical determining factor for cellular responses to IL-17 stimulation in vivo. Future studies are required to delineate additional under-recognized pathophysiological impacts of inflammation-mediated copper mobilization.

## Methods

**Animals**. All animal experiments were conducted in accordance with IACUC guidelines at the Cleveland Clinic Lerner Research Institute. Generation of Steap4 knockout mice (St4 KO) has been described in previous studies[51,52]. Steap4$^{floxed/floxed}$ mice were generated on the C57BL/6 background at Cyagen Biosciences Inc. (Santa Clara, California) using homologous recombination. CDX2-CreERT2 mice (Stock No: 022390) were purchased from the Jackson Laboratory. NOD/scid gamma (NSG) mice

were supplied by the Biological Resource Unit at the Cleveland Clinic from a colony maintained in-house. To generate CDX2-CreERT2 Steap4$^{floxed/floxed}$(Steap4$^{CKO}$), Steap4$^{floxed/floxed}$ mice was bred onto CDX2-CreERT2 background and gender-matched littermates were used for all experiments.

**Generation of inducible, knockdown and knockout cell lines**. The colon cancer cell line LS174T was used to generate all engineered clones in this study. To generate the STEAP4-inducible clone, Flag-tagged human STEAP4 cDNA under the control of Tet-responsive element was transduced into the Ls174t cells along with tetracycline transactivator (tTA) cDNA using lentiviral infection. The infected cells were dispersed into single cells and derived into independent clones. The clones are validated for doxycycline-dependent expression of Flag-tagged STEAP4 with western blot and five independently validated clones were pooled for subsequent experiments. To generate STEAP4 and XIAP knockout cells, guide RNA sequences from the GeCKO (v2) library were used[53]. Specifically, 5′-TTGCATCCAGTGCTCCTGAC-3′ and 5′-GCAGTAGGGTGGTCTTCTGG-3′ were used to knockout STEAP4; 5′-GCATCAACACTGGCACGAGC-3′ was used to knockout XIAP. These guide RNA sequences were cloned into the pLenti-Crispr v2 vector[54,55] and transduced into Ls174t cells by lentiviral infection. The infected cells were dispersed into single cell to generate independent clones and the clones were validated by capillary sequencing of their genomic DNA. Five independently validated clones were pooled and knockout efficiency was further validated by western blot (Supplementary Figs. 1a and 4d). Control cells were generated by transducing Ls174t cells with the same pLenti-Crispr that carried no guide RNA. To generate Ctr1 knockdown cells, control and Ctr1-targeting shRNAs were delivered to the Ls174t cells by lentiviral infection. Infected cells were cultured in selective medium for 48 h to enrich for transduced cells before being subjected to treatment and analysis.

**Mouse and human organoid culture**. Mouse colon organoids were derived from freshly isolated colonic crypts from wild-type or Steap4 knockout mice (St4 KO). Freshly harvested colon tissue was minced and incubated in chelation buffer (5.6 mM Na$_2$HPO$_4$, 8.0 mM KH$_2$PO$_4$, 96.2 mM NaCl, 1.6 mM KCl, 43.4 mM Sucrose, 54.9 mM D-Sorbitol, 2 mM Ethylenediaminetetraacetic acid (EDTA) 1:500 and 0.5 mM Dithiothreitol (DTT)) on ice for 40 min, followed by mechanical dissociation. Released crypts were filtered through a 100 μm cell strainer and precipitated at $300 \times g$ for 5 min at 4 °C. Organoids were cultured in Matrigel (Corning, USA) under organoid culture medium containing 50% advanced DMEM/F12 (Gibco, USA) and 50% of conditioned medium from L-WNR cells, supplemented with 10 μM Y27632 (Caymen Chemicals), 10 μM SB431542 (Caymen Chemicals), 10% fetal bovine serum (FBS) and 2 mM glutamine. Deidentified, discarded cancer tissue that was removed from colorectal cancer patient were used to derive colon cancer organoids. Human colon cancer organoids were established according to published protocol[56] with minor modification. Briefly, fresh colon tumor specimen was dissociated using an enzymatic cocktail consisting of 1.5 mg/mL collagenase VI (Worthington Biochemical, USA), 20 mg/mL hyaluronidase (Worthington Biochemical), 1 mg/mL DNase (Worthington Biochemical) and 10 μM Y27632 (Caymen Chemicals) in advanced DMEM/F12 (Gibco). Dissociated single cells were seeded into Matrigel and maintained in epidermal growth factor-supplemented cancer organoid culture medium derived from the L-WNR cell line[56–58]. Tissue collection and experiments involving clinical samples were approved by the Cleveland Clinic Institutional Review Board (IRB#4134).

**AOM/DSS-induced colon tumorigenesis**. To induce Cre recombinase activity in the colonic epithelial cells, tamoxifen was injected twice per week into 6-week-old Steap4$^{floxed/floxed}$CDX2-CreERT2 and littermate control mice prior to the AOM-DSS treatment for 4 weeks. All mice regardless of their genotypes received intraperitoneal (i.p.) injection of tamoxifen (100 mg/kg) every 2 days for a total of 2 weeks. After tamoxifen treatment, 8-week-old mice were administered with azoxymethane (AOM, Sigma-Aldrich) dissolved in 0.9% NaCl at a dose of

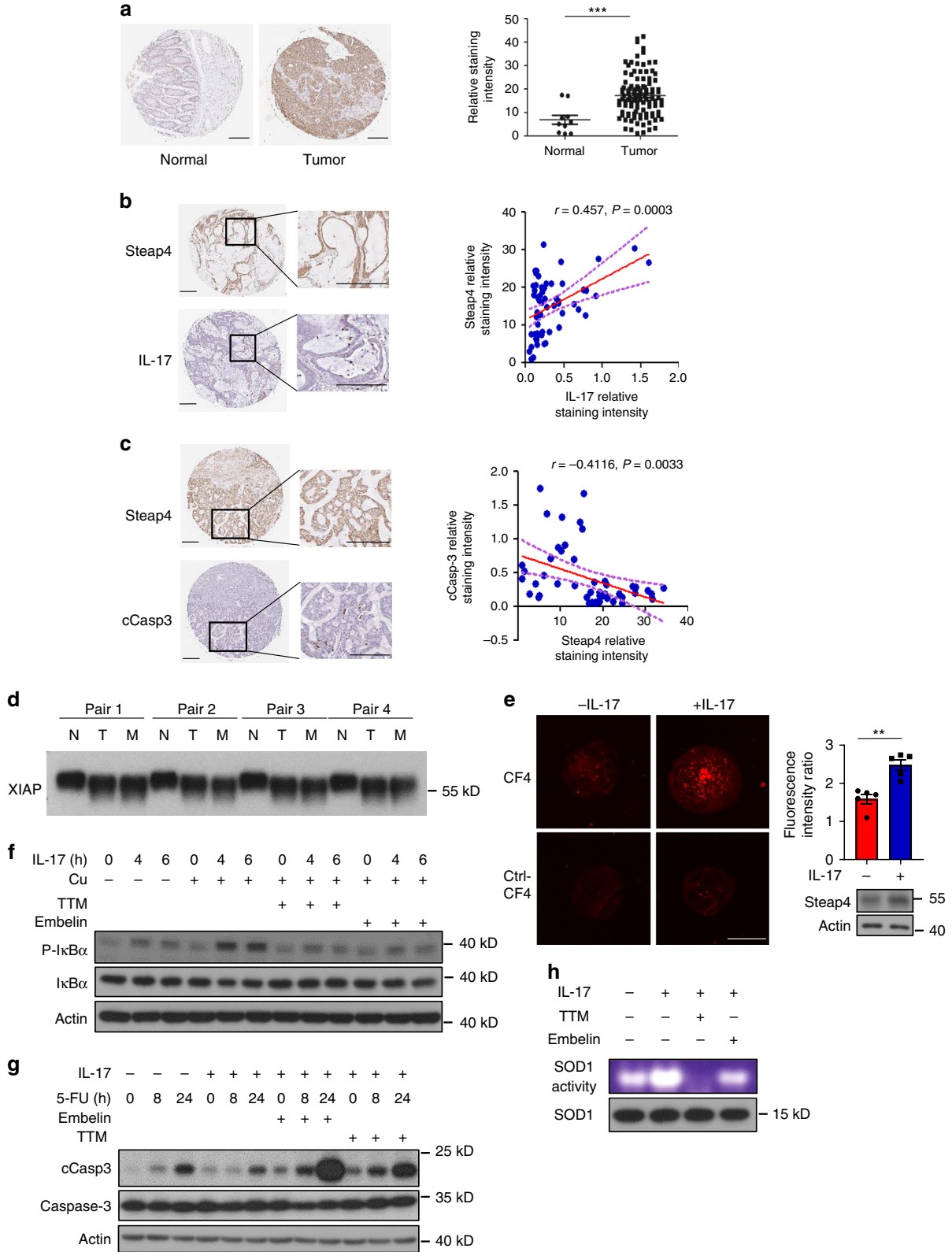

12.5 mg/kg body weight. Subsequently, the mice were treated with 3% dextran sulfate sodium (DSS, MP Biomedicals) in drinking water for 5 days, 1 week after azoxymethane injection, and then with regular water for 16 days. After three cycles of DSS water treatment, tumor-bearing mice were sacrificed. Colon tumor numbers were manually counted and tumor diameters were measured with a caliber. Typical tumors were paraffin-embedded and sectioned at 5 μm thickness for slides. Histology analysis was carried out on hematoxylin-eosin (H&E)-stained or unstained

tumor sections. For the copper chelation experiment, TTM was given at 10 mg/kg body weight per mouse through intraperitoneal injection 3 days a week until the endpoint. Normal saline (0.9% NaCl) was given to the control group.

For IL-17 neutralization experiment, mice were first put on 3% DSS water and antibody injection was initiated 2 days later. In vivo anti-mouse IL-17 (Bio X Cell, Cat# BE0173) or anti-mouse IgG (Bio X Cell, Cat# BE0093) were given to mice through i.p. injection every 3 days until endpoint with 200 μg/mouse dosage.

**Fig. 6 IL-17-induced copper uptake contributes to chemoresistance in human colon cancer. a** Immunostaining for STEAP4 expression in human normal colon and tumor colon tissue. Representative staining is shown. Scale bar, 200 μm. n = 117 biologically independent tissues from different patients. ***P = 0.0007 by two-tailed Student's t test. **b**, **c** STEAP4, IL-17 and cleaved Caspase 3 (cCasp3) were stained on consecutive sections of the same case. Staining intensity of IL-17 and cCasp3 was plotted as a function of STEAP4 intensity. Linear regression was performed to derive the trend line, r and P value. Dotted curve indicates 90% confidence interval. Scale bar, 200 μm. n = 117 biologically independent tissues from different patients. **d** Analysis of XIAP shift on tissue lysates from paired normal (N), tumor (T) and metastatic tissue (M) of the same colon cancer cases. **e** IL-17-induced copper uptake measured by fluorescence probe in colon cancer organoids. n = 5 biologically independent cells. **P = 0.0015 by unpaired two-tailed t test. Scale bar, 100 μm. **f**, **g** Copper and XIAP are required for sustaining IL-17-induced NF-κB activation (**f**) and suppression of 5-FU-induced caspase-3 cleavage (**g**). **h** In-gel SOD1 activity assay on human colon cancer organoids. All data were presented as mean ± SEM. Experiments in (**a**–**c**) were performed once over 117 cases (tissue microarray). Other data represent three independent experiments with similar results.

**Xenograft tumor model**. To investigate tumor growth difference, each flank of NSG mouse was subcutaneously implanted with 1 million engineered LS174T cells in 100 μL cold phosphate-buffered saline (PBS)−Matrigel (Corning) mixture (1:1 ratio). Implanted mice were randomly assigned to each treatment group. Steap4 expression was continuously induced by Doxycycline in drinking water (2 mg/mL with 1% sucrose). TTM (10 mg/kg, Sigma), S63845 (Mcl-1 inhibitor, 12.5 mg/kg, Cayman, Cat# 21131) and BMS-345541[33,34] (IKKα/β inhibitor,125 mg/kg, Sigma, Cat# B9935) were administered through intraperitoneal injection every other day after implantation. Normal saline was injected as the control group. Tumor size was measured using caliber and was initiated 3 days after implantation. Tumor volume was calculated as length/2 × (width)².

**Intrasplenic injection model**. The metastatic potential of colon cancer cells was evaluated using the intrasplenic injection-based xenograft model. Engineered LS174T cells ($1 \times 10^6$/mouse) were surgically injected into the spleens of NSG mice. The spleens of host mice were removed after injection to avoid premature mortality from excessive tumor growth in the spleen. All of the injected cells were transduced with firefly luciferase under the control of a CMV-promoter for in vivo bioluminescence imaging. To examine the impact of STEAP4 expression on the metastatic capacity, STEAP4-inducible cells were grown in the presence or absence of doxycycline for 48 h before injection, resulting in durable expression of exogenous STEAP4 in the xenograft (Supplementary Fig. 7a). Therapeutic evaluation of TTM and 5-FU (Sigma-Aldrich) was performed on NSG mice bearing established xenografts derived from STEAP4-overexpressing LS174T cells. Treatment was initiated 1 week after intrasplenic injection. TTM (10 mg/kg) and/or 5-FU (25 mg/kg) was dissolved in normal saline and administered via intraperitoneal injection on a weekly basis. Normal saline was used as a control. In vivo bioluminescence imaging was used to quantify the amount of colon cancer liver metastasis 30 days after the initial injection of cancer cells prior to necropsy. Mice were anesthetized, injected with D-luciferin (3 mg/20 g body weight, Promega) and imaged under IVIS Spectrum in vivo imaging system (PerkinElmer Inc.).

**Western blot and immunoprecipitation**. Briefly, cells were washed with ice-cold PBS three times and lysed in buffer (1% Triton X-100, 50 mM Tris-HCl pH 7.4, 150 mM NaCl, 12.5 mM β-glycerophosphate, 1.5 mM MgCl₂, 10 mM NaF, 2 mM DTT, 2 mM sodium orthovanadate, 2 mM Ethylene glycol-bis(2-aminoethylether)-N,N,N′,N′-tetraacetic acid (EGTA), and Protease Inhibitor Cocktail (Roche)). Cell extracts were centrifuged at 12,000 × g for 10 min at 4 °C and the supernatants were denatured in Laemmli buffer at 95 °C for 10 min. Samples are resolved by sodium dodecyl sulfate−polyacrylamide gel electrophoresis (SDS-PAGE) and transferred to polyvinylidene fluoride membrane. Subsequently, the membrane was blocked with 5% nonfat milk for 1 h at room temperature and then probed with primary antibody at 4 °C overnight, followed by incubation with HRP-conjugated secondary antibody at room temperature for 1 h. Bands were visualized by chemiluminescence with Enhanced Chemiluminescence. For immunoprecipitation, cell lysates were incubated overnight at 4 °C with antibody plus protein A/G Sepharose beads (GE Healthcare Life Sciences), followed by extensive washes with lysis buffer. Precipitates were eluted by SDS loading buffer and analyzed by the western blot. The following antibodies are used in western blot: STEAP4 (Proteintech 11944-1-AP, 1:500); DYKDDDDK (Cell signaling 14793S, 1:1000); SOD1 (Proteintech 10269-1-AP, 1:1000); SOD1 (Santa Cruze sc-271014, 1:1000); Cleaved Caspase-3 (Cell signaling 9661S, 1:1000); Caspase-3, (Cell Signaling 9662, 1:1000); Ctr1 (Cell signaling 13086S, 1:1000); ATP7A (Thermofisher PA5-36558, 1:1000); ATP7B (Santa Cruz sc-373964, 1:1000); CCS (Santa Cruz sc-55561, 1:1000); p-IκBα (Cell signaling 2859S, 1:1000); p-ERK1/2 (Cell signaling 4370S, 1:1000); p-JNK1/2 (Cell signaling 9255S, 1:1000); P65 (Cell signaling 8242, 1:1000); p-P38 (Cell signaling 4511S, 1:1000); Act1 (eBioscience 14-4040-80, 1:1000); XIAP (Cell signaling 14334S, 1:1000); Lamin (Santa Cruz sc-376248, 1:1000); Tubulin (Cell signaling 2144S, 1:1000).

**XIAP shift assay**. To detect the impact of copper on the electrophoretic mobility of XIAP, treated cells were lysed in buffer containing 25 mM HEPES, 100 mM NaCl, 10% glycerol, and 1% Triton X-100, and EDTA-free protease inhibitor cocktail (Roche). The lysed samples were then resolved in NuPAGE™ 4–12%

Bis-Tris Protein Gels (Invitrogen), transferred to nitrocellulose membranes and then analyzed following regular western blot procedures using anti-XIAP (Cell Signaling Technology, Cat# 2042).

**Nuclear fractionation assay**. Nuclear extraction was performed based on manufacturer's protocol using Nuclei isolation kit (Sigma, Cat# DUC-101). Briefly, cells from 10 cm dish were harvested in 1 mL of ice-cold Nuclei EZ lysis buffer. Cell lysates were then transferred to 1 mL Eppendorf tube and was set on ice for 5 min. Nuclei was collected by centrifugation at 500 × g for 5 min at 4°. After the centrifugation, supernatant was aspirated to a new tube as cytosol fraction for future use. The pellet was washed again in 1 mL Nuclei EZ buffer and then lysed directly in SDS loading buffer for Western blots analysis.

**SOD activity assay**. Cells or organoids were first primed with IL-17 (50 ng/mL) for 12 h, followed by TTM treatment for 12 h and then medium was changed with DMEM plus 10% FBS to allow the SOD activity to recover. Treated cells were harvested and pelleted at 4 °C. Cell pellets were lysed in SOD lysis buffer (10 mM NaPi pH 7.8, 0.1% Triton X-100, 5 mM EDTA, 5 mM EGTA, 50 mM NaCl, 10% Glycerol) plus protease inhibitors. Samples were then resolved on a 10% native gel and then immersed in SOD staining solution (47.6 mL, 1 M K₂HPO₄, 4.8 mL 1 M KH₂PO₄, 140 mg nitro blue tetrazolium, 105 mg Riboflavin in 1050 mL dH₂O) plus 35 μL TEMED at room temperature for 1 h. Stained gel was washed with ddH₂O and scanned at 300dpi.

**Live cell/organoids copper imaging**. Copper sensor compound CF-4 and its control probe ctrl-CF4 were provided by Prof. Christopher J. Chang (University of California, Berkeley). Characterization of the probe has been reported in previous publications[59,60]. For live imaging, cells or organoids were seeded in Nunc™ glass-bottomed dish (Thermofisher) and stimulated in regular culture medium. Treated cells were rinsed twice with PBS and then incubated in phenol red free DMEM supplemented with 10 μM CF4 or control probe for 20 min. After incubation, cells were washed with warm PBS on the dish and then covered with warm phenol-free DMEM for imaging. Confocal microscopy was performed in a CO₂ chamber at 37 °C. Images were obtained on a Leica TCS SP8 laser scanning microscope at ×40 higher magnification with He-Ne laser and HyD hybrid detector (570−700 nm). Images were analyzed using Image-Pro 7.0.

**Atomic absorption spectroscopy**. To measure copper level in culture cells, whole-cell extracts were prepared from a minimum of 5 million cells using 1% SDS buffer and rigorous sonication. Total protein concentration of cell lysates was measured for each sample using Bradford Assay. The protein concentrations were later used as normalizing factor for total copper and iron measurement. Cell lysates were dissolved in nitric acid at 60 °C for 4 h and immediately subjected to atomic absorption spectroscopy (AAS). A flame or graphite furnace AAS instrument was used to determine the total copper abundance in the sample. Concentration standards prepared with CuCl₂ or Fe₂(SO₄)₃ in nitric acid was used to calibrate the standard curve for every experiment.

To determine copper and iron level in mitochondria and cytoplasm, tissue or cell mitochondria were isolated using the Mitochondria Isolation Kits from Thermofisher (Cat# 89874 and Cat# 89801) according to the manufacturer's instruction. Cytosolic fraction, prepared as a byproduct, was saved for copper and iron quantification. For determination of copper and iron content, total mitochondria pellets or 500 mL of cytoplasmic fraction was dissolved or diluted in nitric acid at 60 °C for 4 h before being subjected to AAS. The total copper and iron level was normalized to total cell number.

**Immunohistochemistry**. Mouse tissue was fixed with 10% formalin overnight and then kept in 70% ethanol at 4 °C until processed into paraffin tissue blocks by AML Laboratories (Saint Augustine, Florida). Paraffin sections were subjected to heat-induced epitope retrieval, as recommended by the antibody manufacturer before staining. Deparaffined, epitope-retrieved sections were blocked with 10% normal donkey serum and then incubated with primary antibody overnight. On the second day, the sections were washed with 0.05% Tween PBS followed by incubation with

secondary antibody and streptavidin-HRP for histochemistry and fluorescence-conjugated secondary antibody. Staining was visualized with HRP-substrate chromogen, DAB (3, 3-diaminobenzidine) (BD Pharmingen) or DAPI. The following primary antibodies were used for staining presented in the study: Ki67 (Cell Signaling Technology 12202, 1:200), Cleaved Caspase 3 (Cell Signaling Technology 9661, 1:300), STEAP4 (Proteintech, 11944-1-AP,1:100), IL-17A (R&D Systems, MAB317, 1:300), DYKDDDDK (Cell Signaling, 14793S, 1:300), Ctr1 (Novus Biologicals NBP2-36573, 1:100), TUNEL assay (Sigma, 11684795910).

**Protein purification and in vitro ubiquitination assay.** Codon optimized full-length human XIAP cDNA was cloned into pET28s vector. Recombinant proteins were expressed in BL21 competent *Escherichia coli* and purified with Ni-NTA agarose (Thermo Fisher Scientific). When indicated, $CuCl_2$ (10 μM/mL) was added to the bacteria suspension prior to IPTG-induction. The purified proteins were further desalted and concentrated with Ultra Centrifugal Filter (Millipore). Purity of the recombinant protein was assessed by Coomassie blue staining (Supplementary Fig. 7b). For in vitro ubiquitination assay, K48-only ubiquitin, E1, E2 (UbcH5B) were purchased from Enzo Lifesciences; cleaved caspase-3 was purchased from R&D Systems. The E3-ligase reaction was carried out in 10 mM Mg/ATP solution with 100 μM K48-only ubiquitin, 5 μM recombinant cleaved Caspase-3, 100 nM E1, 1 μM E2 enzyme (UbcH5B) and 1 μM recombinant XIAP.

**Quantification and statistical analysis.** Statistical significance was determined using two-tailed Student's *t* test when comparing the mean of two groups. No adjustment was made for analyses based on Student's *t* test. For growth kinetics analysis, two-way ANOVA was applied to evaluate the significance between treatment groups. Pearson correlation was used to analyze the correlation between IL-17 and STEAP4 expression, between STEAP4 and cleaved Caspase-3. All data were presented as the mean ± standard error of mean (SEM). All tests were two-sided and *P* value less than 0.05 was considered to be statistically significant. All the image quantifications were done by Image-Pro (version 7.0). All the statistics were performed on Graphpad Prism 8.0.

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

## Acknowledgements

We thank Dr. Valeria C. Culotta and Dr. Edward M. Culbertson from Johns Hopkins University for help with in-gel SOD1 activity assay. We thank Dr. Wen Qian, Mr. Jianxin Xiao and Mrs. Jing Ma from Cleveland Clinic for technical support. This work was supported by the National Institutes of Health grants (P01CA062220 and P01 HL103453 to X.L., R01CA193359 to M.F.K., GM 79465 to C.J.C. U01 CA 214300 and R01 237304 to E.H. and S10-OD019972 to the Imaging Core at the Lerner Institute at Cleveland Clinic). C.J.C. is an Investigator with the Howard Hughes Medical Institute.

## Author contributions

X.L. and J.Z. conceived the study and analyzed all the experiments. Y.L. performed majority of the experiments. K.B., F.T., X.C., G.C. assisted in the execution of multiple experiments. S.J. and C.J.C. provided the copper probe. M.F.K. provided human colorectal cancer organoid. P.L.F., T.T.P, E.H., M.W.J., S.B. and G.C.S. gave consultation for experiment design. G.R.S. and C.J.C contributed to the revision of the manuscript. X.L. and J.Z. wrote the paper with input from all coauthors. All the authors read and approved the manuscript.

## Competing interests

The authors declare no competing interests.
