## [Peer Review File · Nature Communications]

Reviewers' comments:

Reviewer #1 (Remarks to the Author):

Summary:

Liao et al. demonstrate here for the first time a connection between extracellular copper (Cu) reduction, inflammatory cell signaling, and colon tumorigenesis. These studies were initiated based on the long held finding that Cu levels are elevated in inflamed tissue and in the tissue and serum of cancer patients. Thus, the authors hypothesized that inflammatory cytokines may contribute to increased Cu levels in the colon to promote colitis and in turn colon cancer. In well-designed experiments, Liao et al. seek to interrogate the contribution of Cu uptake to inflammatory cytokine signaling in colon cancer through the utilization of both in vitro and in vivo perturbations in Cu uptake by genetic and pharmacological means. The authors generated loss- and gain-of-function systems against the Cu and Fe metalloreductase STEAP4, which suggests that Cu uptake is required for colon tumorigenesis and sufficient to increase metastasis. Molecularly, the increased STEAP4 expression in response to IL-17 sustains NFK-B and blocks CASPASE-3-mediated apoptosis through increased activity of the E3 ubiquitin ligase XIAP1. Together, these data support the notion that targeting enhanced Cu uptake during inflammation induced STEAP4-mediated inhibition of apoptosis through XIAP1 may be a treatment option for colon tumorigenesis. While the paper is experimentally solid, the tools are new to the Cu field, and the findings could have clinical impact, there are additional experiments that would improve the manuscript as submitted that warrant revision.

Major and Minor Questions:

1. The authors provide data in Figure 1B that suggests that STEAP4 and CTR1 may colocalize along the basolateral membrane in colonic organoids based on immunofluorescence staining of the adherens junction marker E-CADHERIN and the authors state that this occurs in an IL-17-induced manner. However, each stained organoid in Figure 1B is imaged from a different vantage point and no apical membrane marker was utilized. For example, in the image on the far right, which is used as evidence for colocalization of STEAP4 and CTR1, you cannot visualize the lumen of the organoid. Therefore, the authors should:
 - a. Perform IF of colon organoids for STEAP4 and CTR1 with additional markers of apical and basolateral membranes to confirm the domain of interactions.
 - b. Perform IF of colon organoids for STEAP4 and CTR1 with or without IL-7 priming to determine whether the localization is inflammatory signaling dependent as suggested in the text. Especially since STEAP4 is expressed in the absence of IL-7 (Fig. 1A).
2. The authors provide data in Figure 2 that suggests that STEAP4 overexpression in colon cancer cells is sufficient increase liver metastasis from the spleen. The authors state that "STEAP4-mediated copper uptake promotes tumor growth". However, the experiment as designed only tests whether STEAP4 overexpression can increase the metastasis and colonization of the liver. While the decrease in CASPASE-3 and lack of change in Ki67 staining of the cells stably expressing STEAP4 suggest a deficiency in apoptosis over elevation of cell proliferation, the authors should:
 - a. Test whether STEAP4 overexpression increases the tumor growth of colon cancer cells implanted subcutaneously to ascertain whether STEAP4 overexpression is indeed sufficient to increase tumor growth
 - b. Test whether blocking anti-apoptotic proteins like the BCL-2 family of proteins can restore apoptosis in the presence of STEAP4 overexpression and reduce tumor growth and liver metastasis to determine how much the proposed mechanism of action contributes.

3. While the authors clearly show that overexpression of STEAP4 contributes to IL-17-mediated reductions in apoptosis, the overexpression of STEAP4 was not sufficient to increase cell proliferation. In contrast, treatment with the Cu chelator tetrathiomolybdate (TTM) reduced Ki67 positivity of colon cancer tumors, while also restoring apoptosis. Therefore, the block in tumorigenesis is likely due to both reduced proliferative output and increased apoptosis. The authors should test this directly by:

- a. Expressing a CASPASE-3 mutant (C163A) that cannot induce apoptosis in the context of TTM treatment.
- b. Expressing a CASPASE-3 mutant that cannot be ubiquitinated by XIAP1 in the context of TTM treatment.

4. The authors have compelling data that IL-17-mediated NFkB activation is sustained by the increase in Cu uptake. However, p65/RELA phosphorylation is not increased in Figure 3 in the presence of excess Cu. The authors should:

- a. Indicated which phosphor-p65 antibody is used in the methods
- b. Test activation of NFkB transcription with luciferase assays and target gene induction.
- c. Test whether inhibition of NFkB blocks STEAP4-mediated tumorigenesis.

5. In Figure 5 and Figure 6 the authors provide data utilizing a novel conditional knockout allele of STEAP4 to address the contribution of its metalloreductase activity plays in colon cancer tumorigenesis in an in vivo colitis induced model. It would be great to determine whether IL-17 is mediating the dependence on STEAP4 in vivo. The authors could test this possibility by:

- a. Treating with a neutralizing antibody against IL-17 and testing STEAP4 expression in the mouse model.
- b. Testing whether IL-17 neutralizing antibody is additive to STEAP4 loss in the mouse model.

Reviewer #2 (Remarks to the Author):

Liao et al present a manuscript entitled ""Inflammation mobilizes copper metabolism to promote colon tumorigenesis via an IL-17-STEAP4-XIAP axis"

They show that IL-17 drives cellular copper uptake, which is a newly discovered phenomenon. Mechanistically, they link it to the induction of metalloreductase STEAP4. Increased levels of copper lead to the activation of XIAP, which helps to increase IL-17A induced NFkB activation and to decrease caspase 3 dependent apoptosis. Finally, in a mouse model of colitis-associated tumorigenesis, they find that IL-17-STEAP4-XIAP axis is important for tumor formation. The fact that IL-17A is needed is known, but again STEAP4-XIAP axis is unexpected and novel. Overall, it seems like a very important paper for the field which should be published pending the addressing of some of the rather minor criticism.

1) Fig1A – does it imply that both "NF-kB" and "STAT3" activating cytokines promote Steap4 expression? For this Figure, some mRNA data with some kind of time course would be also important to conclude on transcriptional regulation or lack of thereof.

2) For the data on Fig 5 full characterization of DSS colitis in the absence of Steap 4 (bodyweight, histology, inflammatory gene expression) also should be provided because in this model colitis clearly regulates tumorigenesis.

3) Fig 6- does Steap4 expression in CRC have any bearing on CRC prognosis, particularly for stage II CRC- this can be analyzed using TCGA colon cancer data?

Point-by-point response

Major and Minor Questions:

1. The authors provide data in Figure 1B that suggests that STEAP4 and CTR1 may co-localize along the basolateral membrane in colonic organoids based on immunofluorescence staining of the adherens junction marker E-CADHERIN and the authors state that this occurs in an IL-17-induced manner. However, each stained organoid in Figure 1B is imaged from a different vantage point and no apical membrane marker was utilized. For example, in the image on the far right, which is used as evidence for colocalization of STEAP4 and CTR1, you cannot visualize the lumen of the organoid. Therefore, the authors should:

- a. Perform IF of colon organoids for STEAP4 and CTR1 with additional markers of apical and basolateral membranes to confirm the domain of interactions.
- b. Perform IF of colon organoids for STEAP4 and CTR1 with or without IL-17 priming to determine whether the localization is inflammatory signaling dependent as suggested in the text. Especially since STEAP4 is expressed in the absence of IL-7 (Fig. 1A).

Following the reviewer's suggestions, we stained for integrin $\alpha 6$, a laminin receptor, to mark the basal surface of the intestinal epithelial cells in untreated and IL-17 stimulated colon organoid (**Fig. 1b**). The data showed that while CTR1 is constitutively expressed in the organoids, the expression of STEAP4 is robustly induced by IL-17. This observation is consistent with our new data indicating that IL-17 neutralization reduces STEAP4 expression in colon tissue from DSS-treated mice (**Supplementary Fig. 1b-d**).

2. The authors provide data in Figure 2 that suggests that STEAP4 overexpression in colon cancer cells is sufficient increase liver metastasis from the spleen. The authors state that "STEAP4-mediated copper uptake promotes tumor growth". However, the experiment as designed only tests whether STEAP4 overexpression can increase the metastasis and colonization of the liver. While the decrease in CASPASE-3 and lack of change in Ki67 staining of the cells stably expressing STEAP4 suggest a deficiency in apoptosis over elevation of cell proliferation, the authors should:

- a. Test whether STEAP4 overexpression increases the tumor growth of colon cancer cells implanted subcutaneously to ascertain whether STEAP4 overexpression is indeed sufficient to increase tumor growth
- b. Test whether blocking anti-apoptotic proteins like the BCL-2 family of proteins can restore apoptosis in the presence of STEAP4 overexpression and reduce tumor growth and liver metastasis to determine how much the proposed mechanism of action contributes.

We analyzed the growth kinetics of xenografts derived from Ls174t cells with and without STEAP4 overexpression. Consistent with what we observed in the intrasplenic injection-mediated model, STEAP4 expression was associated with faster tumor growth and reduced caspase 3 activation, which was reversed by TTM treatment (**Supplementary Fig. 2a-d**). The new data also indicated that STEAP4 expression promotes tumor growth.

To assess the contribution of STEAP4-mediated suppression of caspase 3 activation to the tumor growth, we used inhibitors against Bcl-2 family proteins as a tool to induce caspase 3 activation. We tested several bioavailable inhibitors and found that Ls174t cells were most sensitive to s63845 (data not shown), which specifically inhibits Mcl-1.

In support of our finding that STEAP4-mediated copper uptake activates XIAP to degrade Caspase 3, STEAP4 overexpression actually dampened s63845-mediated Caspase 3 activation at low concentrations (~100nM). In vivo, although s63845 treatment suppressed the growth of both STEAP-expressing (Dox+) and control (Dox-) xenografts, the s63845-treated STEAP4-expressing (Dox+) xenografts showed faster growth compared to control tumors treated with s63845 (**Supplementary Fig. 4b-c**). Consistently, immunofluorescence staining revealed that while s63845 treatment increased caspase 3 activation in both STEAP4 expressing xenograft tissue and control tumor tissue, STEAP4 expression was able to reduce the frequencies of caspase 3+ cells in

s63845 treated tumor (**Supplementary Fig. 4d**). Taken together, the data suggested that STEAP4 expression-mediated suppression of Caspase 3 activity contributes majorly to the growth advantage in the xenograft model.

3. While the authors clearly show that overexpression of STEAP4 contributes to IL-17-mediated reductions in apoptosis, the overexpression of STEAP4 was not sufficient to increase cell proliferation. In contrast, treatment with the Cu chelator tetrathiomolybdate (TTM) reduced Ki67 positivity of colon cancer tumors, while also restoring apoptosis. Therefore, the block in tumorigenesis is likely due to both reduced proliferative output and increased apoptosis. The authors should test this directly by:

a. Expressing a CASPASE-3 mutant (C163A) that cannot induce apoptosis in the context of TTM treatment.

The reviewer insightfully pointed out that Cu also plays a role in sustaining cell proliferation, which has been elegantly demonstrated by previous studies (Brady et al., 2017; Brady et al., 2014). The baseline frequency of proliferative cells was over 70% (**Fig. 2h**). This high baseline proliferation might have masked the proliferative impact of STEAP4. Nevertheless, we revisited our Ki67 staining from the intrasplenic injection model. A more extensive survey of tissue indeed suggested a trend of increased Ki67 in STEAP4+ tumor. The baseline frequency in the flank xenograft model was comparatively lower to that in the liver metastases. STEAP4 expression led to a statistically significant increase in Ki67+ cells in the xenograft tumors implanted in the flank. Importantly, the increase in Ki67 cells was suppressed by TTM treatment (**Supplementary Fig. 2d**). This result suggested that indeed the block in tumorigenesis by TTM is likely due to a combination of reduced proliferative output and increased apoptosis.

Furthermore, following the reviewer's suggestion, we generated an Ls174t cell line that express an enzymatically defective caspase 3 mutant (C163A), which conferred resistance to TTM-induced apoptosis as revealed by TUNEL assay (**Supplementary Fig. 5c**). Despite the resistance to TTM-induced apoptosis, TTM treatment was able to reduce the frequency of BrdU+ cells both in cell culture (**Supplementary Fig. 5a**) and suppress the overall tumor growth in vivo (**Supplementary Fig. 5d-e**). Taken together, the data indicate that copper chelation-mediated reduction in cell proliferation contributed to blunted tumor growth.

b. Expressing a CASPASE-3 mutant that cannot be ubiquitinated by XIAP in the context of TTM treatment. This experiment suggested by the reviewer sought to uncouple the regulation of Caspase 3 from XIAP. Based on our data and proposed model of mechanism, it would follow that a caspase 3 mutant resistant to XIAP-mediated degradation would partially abrogate the chemoresistance conferred by STEAP4 overexpression. Since the precise lysine residue in Caspase 3 that XIAP ubiquitinates remains to be identified, we tested a previously defined mutant Caspase-3 mutant (F381H) that do not interact with XIAP to decouple it from STEAP4-Cu-mediated negative regulation (**Supplementary Fig. 5h**) (Riedl et al., 2001). Unfortunately, while we can transiently express this mutant we were unable to obtain stable expression. We therefore only tested the impact of decoupling Caspase 3 from XIAP in cell culture model. Consistent with our hypothesis, we detected increased TUNEL activity at the baseline in cells STEAP4-expressing cells that carry the mutant caspase 3 (**Supplementary Fig. 5i**). Furthermore, while TTM treatment increased the TUNEL activity in IL-17-treated STEAP4 expressing cells with wild-type caspase 3, it failed to further enhance cell death in cells that expressed mutant caspase 3 (**Supplementary Fig. 5i**). Taken together, the data indicate that STEAP4-mediated suppression of caspase 3 activation was dependent the XIAP-caspase 3 interaction. Collectively, the data strengthen our previous conclusion on a crucial contribution of STEAP4-mediated Cu-dependent negative regulation of caspase 3 activity by XIAP in tumor growth.

4. The authors have compelling data that IL-17-mediated NFKB activation is sustained by the increase in Cu uptake. However, p65/RELA phosphorylation is not increased in Figure 3 in the presence of excess Cu. The authors should:

- a. Indicated which phosphor-p65 antibody is used in the methods
- b. Test activation of NFKB transcription with luciferase assays and target gene induction.
- c. Test whether inhibition of NFKB blocks STEAP4-mediated tumorigenesis.

We thank the reviewer for the suggestions. Upon reviewing the original films, we observed small but noticeable increased phospho-p65 in response to STEAP4-mediated copper uptake. Given the high level of signal from the western blots, we believe that the p-P65 blots were all overexposed, diminishing the dynamic range of detection. To unequivocally determine the impact of STEAP4-mediated copper uptake on IL-17-induced NF- κ B activation. We performed the following experiments:

1. We performed NF κ B luciferase reporter assay in the presence or absence of STEAP4 expression in response to IL-17 stimulation in HeLa cell line. Induction of STEAP4 enhanced IL-17-induced NF κ B activity (**Supplementary Fig. 1f**).

2. We performed nuclear fractionation in IL-17-stimulated wild-type and STEAP4 knockout cells and analyzed the abundance of p65. The data clearly showed that STEAP4-deficiency reduced the amount of nuclear p65 after 4 and 6 hours of IL-17 stimulation (**Fig. 3c-d**). The data collectively indicated that IL-17-mediated NF κ B activation is sustained by STEAP4-mediated the increase in Cu uptake.

Data from these new experiments clearly showed that STEAP4-mediated copper uptake sustained IL-17-induced NF κ B activation. We therefore used NF κ B nuclear translocation and luciferase assay as readouts for NF κ B activation instead of phospho-P65 blots.

3. XIAP is known to activate NF κ B via the IKK α/β kinases. Next, we examined the impact of STEAP4-mediated NF- κ B activation on tumor growth using a selective IKK α/β (Burke et al., 2003; Enzler et al., 2011). Consistent with the sustained IL-17-induced NF κ B activation in response to STEAP4-overexpression, pharmacological inhibition of NF κ B activation blunted STEAP4-induced cell proliferation in IL-17-stimulated Ls174t cells (**Supplementary Fig. 4e**). In line with the *in vitro* data, inhibition of NF κ B activation reduced tumor cell proliferation and attenuated the growth advantage conferred by STEAP4 expression in the xenograft model (**Supplementary Fig. 4f-g**). However, inhibition of NF κ B did not normalize the growth kinetics (**Supplementary Fig. 4f-g**). The residual tumor growth advantage was likely contributed by the STEAP4-mediated suppression of caspase 3 activation. Collectively, the data indicated that STEAP4-sustained NF κ B activation possibly contributed to tumor growth by promoting cell proliferation.

5. In Figure 5 and Figure 6 the authors provide data utilizing a novel conditional knockout allele of STEAP4 to address the contribution of its metalloredutase activity plays in colon cancer tumorigenesis in an *in vivo* colitis induced model. It would be great to determine whether IL-17 is mediating the dependence on STEAP4 *in vivo*. The authors could test this possibility by:

a. Treating with a neutralizing antibody against IL-17 and testing STEAP4 expression in the mouse model.

We analyzed STEAP4 expression in lysates from colons of DSS-treated mice receiving either control IgG or anti-IL-17 neutralizing antibodies. IL-17 neutralization dramatically suppressed STEAP4 expression in the inflamed colon (**Supple. Fig. 1b**).

b. Testing whether IL-17 neutralizing antibody is additive to STEAP4 loss in the mouse model.

We attempted to perform IL-17 neutralization on intestinal epithelial cells specific STEAP4 knockout mice and littermate controls as the reviewers suggested. Unfortunately, due to the protective role of IL-17 on the integrity of intestinal epithelium (Lee et al., 2015; Maxwell et al., 2015)(**Supplemental Fig. 1b-d**), chronic neutralization of IL-17 led to over 50% lethality after the 2nd cycle of DSS challenge. Lowering of DSS concentration failed to induce sufficient chronic inflammation and tumorigenesis. As a result, we were unable to attain statistically sound dataset despite multiple attempts. Thus, we can only speculate on possible outcomes. We have previously reported that IL-17-induces a group of Plet1+ progenitor cells from the Lgr5+ stem cells in the colon (Zepp et al., 2017), which critically contribute to tissue repair and tumorigenesis. Given the versatile role of IL-17, STEAP4 is mostly likely one of the effector molecules of IL-17 in colon tumorigenesis. Therefore, it is possible that IL-17 neutralization may have additive effect on tumor reduction in epithelial cell specific STEAP4 knockout mice.

Reviewer #2 (Remarks to the Author):

Liao et al present a manuscript entitled ""Inflammation mobilizes copper metabolism to promote colon tumorigenesis via an IL-17-STEAP4-XIAP axis"

They show that IL-17 drives cellular copper uptake, which is a newly discovered phenomenon. Mechanistically, they link it to the induction of metalloreductase STEAP4. Increased levels of copper lead to the activation of XIAP, which helps to increase IL-17A induced NFkB activation and to decrease caspase 3 dependent apoptosis. Finally, in a mouse model of colitis-associated tumorigenesis, they find that IL-17-STEAP4-XIAP axis is important for tumor formation. The fact that IL-17A is needed is known, but again STEAP4-XIAP4 axis is unexpected and novel. Overall, it seems like a very important paper for the field which should be published pending the addressing of some of the rather minor criticism.

1) Fig1A – does it imply that both “NF-kB” and “STAT3” activating cytokines promote Steap4 expression? For this Figure, some mRNA data with some kind of time course would be also important to conclude on transcriptional regulation or lack of thereof.

We completely agree with the reviewer’s observation regarding the roles of NFkB and STAT3 in the induction of STEAP4. Analysis of STEAP4 promoter sequence indeed have revealed binding sites for both NFkB and STAT3. To further determine whether STEAP4 was transcriptionally upregulated by IL-17 and IL-22 (STAT3 activator). We performed real-time analysis and found that STEAP4 was induced within 4 hour of IL-17 and IL-22 stimulation (**Supplementary Fig. 1a**). Furthermore, blockade of transcription using actinomycin D completely abrogated IL-17 and IL-22 induced STEAP4 transcripts (**Supplementary Fig. 1a**). Collectively, the data supports that STEAP4 expression is transcriptionally regulated by NF-κB and STAT3 activators.

[redacted]

In sum, our data suggested that IL-17 regulates STEAP4 expression at both transcriptional and post-transcriptional level. However, the post-transcriptional regulation of STEAP4 expression is out of the scope of the current study. Additional studies are required to investigate the impact of IL-17 on the metabolism of STEAP4 transcripts.

2) For the data on Fig 5 full characterization of DSS colitis in the absence of Steap 4 (bodyweight, histology, inflammatory gene expression) also should be provided because in this model colitis clearly regulates tumorigenesis.

Consistent with the protective role of IL-17 signaling in colitis, STEAP4 deficiency in the intestinal epithelial cells led to an exacerbated weight loss increased gut permeability and reduced epithelial cell proliferation in response to DSS challenge (**Supplementary Fig. 6a-d**). This data further support a role for IL-17-induced STEAP4-mediated cell proliferation in colitis-associated tumorigenesis.

3) Fig 6- does Steap4 expression in CRC have any bearing on CRC prognosis, particularly for stage II CRC- this can be analyzed using TCGA colon cancer data?

Higher STEAP4 expression correlated with worse overall survival in the TCGA dataset (**Supplementary Fig. 6e**, $p < 0.05$ by log-rank test). This trend is preserved when looking at only STAGE II CRC but the data did not reach statistical significance. Nevertheless, this data warrants future studies using immunohistochemistry to evaluate the predictive value of STEAP4 expression in stage II CRC.

References:

- Brady, D.C., M.S. Crowe, D.N. Greenberg, and C.M. Counter. 2017. Copper Chelation Inhibits BRAF(V600E)-Driven Melanomagenesis and Counters Resistance to BRAF(V600E) and MEK1/2 Inhibitors. *Cancer Res* 77:6240-6252.
- Brady, D.C., M.S. Crowe, M.L. Turski, G.A. Hobbs, X. Yao, A. Chaikuad, S. Knapp, K. Xiao, S.L. Campbell, D.J. Thiele, and C.M. Counter. 2014. Copper is required for oncogenic BRAF signalling and tumorigenesis. *Nature* 509:492-496.
- Burke, J.R., M.A. Pattoli, K.R. Gregor, P.J. Brassil, J.F. MacMaster, K.W. McIntyre, X. Yang, V.S. Iotzova, W. Clarke, J. Strnad, Y. Qiu, and F.C. Zusi. 2003. BMS-345541 is a highly selective inhibitor of I kappa B kinase that binds at an allosteric site of the enzyme and blocks NF-kappa B-dependent transcription in mice. *J Biol Chem* 278:1450-1456.
- Enzler, T., Y. Sano, M.K. Choo, H.B. Cottam, M. Karin, H. Tsao, and J.M. Park. 2011. Cell-selective inhibition of NF-kappaB signaling improves therapeutic index in a melanoma chemotherapy model. *Cancer discovery* 1:496-507.
- Lee, J.S., C.M. Tato, B. Joyce-Shaikh, M.F. Gulen, C. Cayatte, Y. Chen, W.M. Blumenschein, M. Judo, G. Ayanoglu, T.K. McClanahan, X. Li, and D.J. Cua. 2015. Interleukin-23-Independent IL-17 Production Regulates Intestinal Epithelial Permeability. *Immunity* 43:727-738.
- Maxwell, J.R., Y. Zhang, W.A. Brown, C.L. Smith, F.R. Byrne, M. Fiorino, E. Stevens, J. Bigler, J.A. Davis, J.B. Rottman, A.L. Budelsky, A. Symons, and J.E. Towne. 2015. Differential Roles for Interleukin-23 and Interleukin-17 in Intestinal Immunoregulation. *Immunity* 43:739-750.
- Riedl, S.J., M. Renatus, R. Schwarzenbacher, Q. Zhou, C. Sun, S.W. Fesik, R.C. Liddington, and G.S. Salvesen. 2001. Structural basis for the inhibition of caspase-3 by XIAP. *Cell* 104:791-800.
- Zepp, J.A., J. Zhao, C. Liu, K. Bulek, L. Wu, X. Chen, Y. Hao, Z. Wang, X. Wang, W. Ouyang, M.F. Kalady, J. Carman, W.P. Yang, J. Zhu, C. Blackburn, Y.H. Huang, T.A. Hamilton, B. Su, and X. Li. 2017. IL-17A-Induced PLET1 Expression Contributes to Tissue Repair and Colon Tumorigenesis. *J Immunol* 199:3849-3857.

REVIEWERS' COMMENTS:

Reviewer #1 (Remarks to the Author):

Inflammation mobilizes copper metabolism to promote colon tumorigenesis via an IL-17-STEAP4-XIAP axis

Summary: The paper is experimentally solid, the tools are new to the Cu field, and the findings could have clinical impact. The additional experiments have significantly improved the manuscript and it warrants publication.

Major and Minor Questions:

1. The authors provide data in Figure 1B that suggests that STEAP4 and CTR1 may colocalize along the basolateral membrane in colonic organoids based on immunofluorescence staining of the adherens junction marker E-CADHERIN and the authors state that this occurs in an IL-17-induced manner. However, each stained organoid in Figure 1B is imaged from a different vantage point and no apical membrane marker was utilized. For example, in the image on the far right, which is used as evidence for colocalization of STEAP4 and CTR1, you cannot visualize the lumen of the organoid. Therefore, the authors should:

- a. Perform IF of colon organoids for STEAP4 and CTR1 with additional markers of apical and basolateral membranes to confirm the domain of interactions.
- b. Perform IF of colon organoids for STEAP4 and CTR1 with or without IL-7 priming to determine whether the localization is inflammatory signaling dependent as suggested in the text. Especially since STEAP4 is expressed in the absence of IL-7 (Fig. 1A).

- In response to the reviewer's comments, the authors have successfully addressed this reviewer's concerns with new data presented in Figure 1B and Supplementary Figure 1B-D.

2. The authors provide data in Figure 2 that suggests that STEAP4 overexpression in colon cancer cells is sufficient increase liver metastasis from the spleen. The authors state that "STEAP4-mediated copper uptake promotes tumor growth". However, the experiment as designed only tests whether STEAP4 overexpression can increase the metastasis and colonization of the liver. While the decrease in CASPASE-3 and lack of change in Ki67 staining of the cells stably expressing STEAP4 suggest a deficiency in apoptosis over elevation of cell proliferation, the authors should:

- a. Test whether STEAP4 overexpression increases the tumor growth of colon cancer cells implanted subcutaneously to ascertain whether STEAP4 overexpression is indeed sufficient to increase tumor growth
- b. Test whether blocking anti-apoptotic proteins like the BCL-2 family of proteins can restore apoptosis in the presence of STEAP4 overexpression and reduce tumor growth and liver metastasis to determine how much the proposed mechanism of action contributes.

- In response to the reviewer's comments, the authors have successfully addressed this reviewer's concerns with new data presented in Supplementary Figure 2A-D and Supplementary Figure 4B-C.

3. While the authors clearly show that overexpression of STEAP4 contributes to IL-17-mediated reductions in apoptosis, the overexpression of STEAP4 was not sufficient to increase cell proliferation. In contrast, treatment with the Cu chelator tetrathiomolybdate (TTM) reduced Ki67 positivity of colon cancer tumors, while also restoring apoptosis. Therefore, the block in tumorigenesis is likely due to both reduced proliferative output and increased apoptosis. The authors should test this directly by:

- a. Expressing a CASPASE-3 mutant (C163A) that cannot induce apoptosis in the context of TTM treatment.
- b. Expressing a CASPASE-3 mutant that cannot be ubiquitinated by XIAP1 in the context of TTM treatment.

- In response to the reviewer's comments, the authors have successfully addressed this reviewer's concerns with new data presented in Supplementary Figure 2D and Supplementary Figure 5A-I.

4. The authors have compelling data that IL-17-mediated NFκB activation is sustained by the increase in Cu uptake. However, p65/RELA phosphorylation is not increased in Figure 3 in the presence of excess Cu. The authors should:

- a. Indicated which phosphor-p65 antibody is used in the methods
- b. Test activation of NFκB transcription with luciferase assays and target gene induction.
- c. Test whether inhibition of NFκB blocks STEAP4-mediated tumorigenesis.

- In response to the reviewer's comments, the authors have successfully addressed this reviewer's concerns with new data presented in Figure 1B and Supplementary Figure 1F, Figure 3D,D, and Supplementary Figure 4E-G.

5. In Figure 5 and Figure 6 the authors provide data utilizing a novel conditional knockout allele of STEAP4 to address the contribution of its metalloredutase activity plays in colon cancer tumorigenesis in an in vivo colitis induced model. It would be great to determine whether IL-17 is mediating the dependence on STEAP4 in vivo. The authors could test this possibility by:

- a. Treating with a neutralizing antibody against IL-17 and testing STEAP4 expression in the mouse model.
- b. Testing whether IL-17 neutralizing antibody is additive to STEAP4 loss in the mouse model.

- In response to the reviewer's comments, the authors have successfully addressed this reviewer's concerns with new data presented in Figure 1B and Supplementary Figure 1B-D, Figure 3D,D, and Supplementary Figure 4E-G.

Reviewer #2 (Remarks to the Author):

I think author reasonably addressed all of the comments and the paper should be accepted

We are very gratified that the reviewers have found that the new data provided in the revision successfully addressed all of the concerns raised in the first round of review.

REVIEWERS' COMMENTS:

Reviewer #1 (Remarks to the Author):

Inflammation mobilizes copper metabolism to promote colon tumorigenesis via an IL-17-STEAP4-XIAP axis

Summary: The paper is experimentally solid, the tools are new to the Cu field, and the findings could have clinical impact. The additional experiments have significantly improved the manuscript and it warrants publication.

Major and Minor Questions:

1. The authors provide data in Figure 1B that suggests that STEAP4 and CTR1 may colocalize along the basolateral membrane in colonic organoids based on immunofluorescence staining of the adherens junction marker E-CADHERIN and the authors state that this occurs in an IL-17-induced manner. However, each stained organoid in Figure 1B is imaged from a different vantage point and no apical membrane marker was utilized. For example, in the image on the far right, which is used as evidence for colocalization of STEAP4 and CTR1, you cannot visualize the lumen of the organoid. Therefore, the authors should:

- a. Perform IF of colon organoids for STEAP4 and CTR1 with additional markers of apical and basolateral membranes to confirm the domain of interactions.
- b. Perform IF of colon organoids for STEAP4 and CTR1 with or without IL-7 priming to determine whether the localization is inflammatory signaling dependent as suggested in the text. Especially since STEAP4 is expressed in the absence of IL-7 (Fig. 1A).

- In response to the reviewer's comments, the authors have successfully addressed this reviewer's concerns with new data presented in Figure 1B and Supplementary Figure 1B-D.

2. The authors provide data in Figure 2 that suggests that STEAP4 overexpression in colon cancer cells is sufficient increase liver metastasis from the spleen. The authors state that "STEAP4-mediated copper uptake promotes tumor growth". However, the experiment as designed only tests whether STEAP4 overexpression can increase the metastasis and colonization of the liver. While the decrease in CASPASE-3 and lack of change in Ki67 staining of the cells stably expressing STEAP4 suggest a deficiency in apoptosis over elevation of cell proliferation, the authors should:

- a. Test whether STEAP4 overexpression increases the tumor growth of colon cancer cells implanted subcutaneously to ascertain whether STEAP4 overexpression is indeed sufficient to increase tumor growth
- b. Test whether blocking anti-apoptotic proteins like the BCL-2 family of proteins can restore apoptosis in the presence of STEAP4 overexpression and reduce tumor growth and liver metastasis to determine how much the proposed mechanism of action contributes.

- In response to the reviewer's comments, the authors have successfully addressed this reviewer's concerns with new data presented in Supplementary Figure 2A-D and Supplementary Figure 4B-C.

3. While the authors clearly show that overexpression of STEAP4 contributes to IL-17-mediated reductions in apoptosis, the overexpression of STEAP4 was not sufficient to increase cell proliferation. In contrast, treatment with the Cu chelator tetrathiomolybdate (TTM) reduced Ki67 positivity of colon cancer tumors, while also restoring apoptosis. Therefore, the block in tumorigenesis is likely due to both reduced proliferative output and increased apoptosis. The authors should test this directly by:

- a. Expressing a CASPASE-3 mutant (C163A) that cannot induce apoptosis in the context of TTM treatment.
- b. Expressing a CASPASE-3 mutant that cannot be ubiquitinated by XIAP1 in the context of TTM treatment.

- In response to the reviewer's comments, the authors have successfully addressed this reviewer's concerns with new data presented in Supplementary Figure 2D and Supplementary Figure 5A-I.

4. The authors have compelling data that IL-17-mediated NF κ B activation is sustained by the increase in Cu uptake. However, p65/RELA phosphorylation is not increased in Figure 3 in the presence of excess Cu. The authors should:

- a. Indicated which phosphor-p65 antibody is used in the methods
- b. Test activation of NF κ B transcription with Luciferase assays and target gene induction.
- c. Test whether inhibition of NF κ B blocks STEAP4-mediated tumorigenesis.

- In response to the reviewer's comments, the authors have successfully addressed this reviewer's concerns with new data presented in Figure 1B and Supplementary Figure 1F, Figure 3D,D, and Supplementary Figure 4E-G.

5. In Figure 5 and Figure 6 the authors provide data utilizing a novel conditional knockout allele of STEAP4 to address the contribution of its metalloredutase activity plays in colon cancer tumorigenesis in an in vivo colitis induced model. It would be great to determine whether IL-17 is mediating the dependence on STEAP4 in vivo. The authors could test this possibility by:

- a. Treating with a neutralizing antibody against IL-17 and testing STEAP4 expression in the mouse model.
- b. Testing whether IL-17 neutralizing antibody is additive to STEAP4 loss in the mouse model.

- In response to the reviewer's comments, the authors have successfully addressed this reviewer's concerns with new data presented in Figure 1B and Supplementary Figure 1B-D, Figure 3D,D, and Supplementary Figure 4E-G.

Reviewer #2 (Remarks to the Author):

I think author reasonably addressed all of the comments and the paper should be accepted